

# MODIS Collection 6 MAIAC Algorithm

Alexei Lyapustin[1], Yujie Wang[2], Sergey Korkin[3], Dong Huang[4]

[1]Laboratory for Atmospheres, NASA Goddard Space Flight Center, Greenbelt, Maryland, USA
[2]University of Maryland Baltimore County, Baltimore, Maryland, USA
[3]Universities Space Research Association, Columbia, Maryland, USA
[4]Science Systems and Applications, Inc., Lanham, MD 20709, USA

*Correspondence to*: Alexei.I.Lyapustin@nasa.gov.

**Abstract.** This paper describes the latest version of algorithm MAIAC used for processing of the MODIS Collection 6 data record. Since initial publication in 2011-2012, MAIAC has changed considerably to adapt global processing and improve

cloud/snow detection, aerosol retrievals and atmospheric correction of MODIS data. The main changes include 1) transition from 25km to 1km scale for retrieval of the spectral regression coefficient (SRC) which helped remove occasional blockiness at 25km scale in the aerosol optical depth (AOD) and in the surface reflectance; 2) continuous improvements of cloud detection; 3) introduction of "Smoke" and "Dust" tests to discriminate absorbing fine and coarse mode aerosols; 4) adding over-water processing; 5) general optimization of the LUT-based radiative transfer for the global processing, and

others. MAIAC provides an inter-disciplinary suite of atmospheric and land products, including: cloud mask (CM), column water vapor (CWV), AOD at 0.47 and 0.55μm, aerosol type (background/smoke/dust), and fine mode fraction over water; spectral bidirectional reflectance factors (BRF), parameters of Ross-Thick Li-Sparse (RTLS) BRDF model and instantaneous albedo; for snow-covered surfaces, we provide sub-pixel snow fraction and snow grain size. All products come in standard HDF4 format at 1km resolution, except BRF which is also provided at 500m resolution, on Sinusoidal grid adopted by the

MODIS land team. All products are provided on per-observation basis in daily files except BRDF/albedo which is reported every 8 days. Because MAIAC uses a time series approach, the BRDF/albedo are naturally gap-filled over land where missing values are filled-in with results from the previous retrieval. While the BRDF model is reported for MODIS 'land' bands 1-7 and 'ocean' band 8, BRF is reported for both 'land' and 'ocean' bands 1-12. This paper focuses on MAIAC cloud detection, aerosol retrievals and atmospheric correction and describes MCD19 data products and quality assurance (QA)

flags.

## 1. Introduction

Simple and fast *swath*-based processing with Lambertian surface model is the basis of the Moderate Resolution Imaging Spectroradiometer (MODIS) Dark Target (DT) (Levy et al., 2013) and VIIRS (Jackson et al., 2013) aerosol retrievals and atmospheric correction (AC) (Vermote and Kotchenova, 2008). In *swath* data, the satellite footprint and its location are orbit-

dependent and change with scan angle making it difficult to characterize the surface BRDF. The above algorithms rely on



prescribed spectral surface reflectance (SR) ratios to make aerosol retrievals. The SR ratios represent statistically average relationships with relatively large variance. When the surface brightness increases and sensitivity of the top of atmosphere (TOA) radiance to aerosols decreases, this lack of accurate knowledge of surface reflectance becomes a major issue.

The Multi-Angle Implementation of Atmospheric Correction (MAIAC) algorithm uses a physical atmosphere-surface model

where the model parameters are defined from measurements (Lyapustin et al., 2011a,b; 2012a,b) with minimal assumptions. Instead of swath-based processing, we start with gridding MODIS L1B measurements to a fixed 1km grid, and with accumulating a time series of data for up to 16-days using a sliding window technique. This allows us to observe the same grid cell over time helping separate atmospheric and surface contributions with the time series analysis and characterize surface bi-directional reflectance distribution function (BRDF) using multi-angle observations from different orbits. Besides

BRDF retrieval, the fixed surface representation (grid) allows us to characterize and store unique surface spectral, spatial, thermal etc. signatures for each 1km grid cell helping to increase accuracy of the entire processing, from cloud and snow detection to aerosol retrievals and atmospheric correction (AC).

Since its introduction in 2011-2012, we significantly changed and improved several key parts of MAIAC, namely cloud and snow detection, characterization of spectral regression coefficient (SRC) and aerosol retrieval, and transformed the algorithm

from regional to global. The intermediate versions of MAIAC were continuously tested by the land and air quality communities using our processing of MODIS data with the NASA Center for Climate Simulations (NCCS) and product release via NCCS ftp portal (ftp://maiac@dataportal.nccs.nasa.gov/DataRelease). Analysis by Hilker et al. (2012; 2014; 2015), Maeda et al. (2016) and others showed a dramatic (up to a factor of 3-5) increase in the accuracy of MAIAC surface reflectance compared to MODIS standard products MOD09, MOD035 over tropical Amazon. Since 2014, all major

studies of Amazon tropical forests, which used MODIS data, relied on MAIAC processing (e.g., Saleska et al., 2016; Lopes et al., 2016; Alden et al., 2016; Guan et al., 2015; Bi et al., 2015, 2016; Jones et al., 2014; Maeda et al., 2017; Wagner et al., 2017). Recently, Chen et al. (2017) reported an improvement in the leaf area index (LAI) retrievals with MODIS LAI/FPAR algorithm when using MAIAC instead of standard MODIS MOD09 input. A good accuracy, high 1km spatial resolution and high retrieval coverage made MAIAC aerosol optical depth (AOD) a focus of numerous air quality studies, e.g.

(Chudnovsky et al., 2013; Kloog et al., 2014; Just et al., 2015; Di et al., 2016; Stafoggia et al., 2016; Tang et al., 2017; Xiao et al., 2017) to name a few. Currently published validation studies (Martins et al., 2017; Superczynski et al, 2017) show a good MAIAC AOD accuracy over American continents, and an improved accuracy and coverage over North America compared to the operational VIIRS algorithm (Superczynski et al, 2017). An emerging comparative aerosol validation analysis over North America (Jethva et al., to be submitted) and South Asia (Mhawish et al., in review) shows that MAIAC

has a comparable or better accuracy than the DT algorithm over dark surfaces and generally improves accuracy over the Deep Blue (DB) algorithm (Hsu et al., 2013) over bright surfaces.

The MAIAC MODIS Collection 6 (with enhanced calibration (C6+) which added polarization correction of MODIS Terra, removed residual trends of both Terra and Aqua, and cross-calibrated Terra to Aqua (Lyapustin et al., 2014)) processing is ongoing on the MODIS Adaptive Processing System (MODAPS). It is expected to be completed in spring of 2018 creating a



new MODIS product MCD19 accessible via Land Product Distributed Active Archive Center (LP DAAC). MAIAC offers an interdisciplinary suite of products for the Land, Atmosphere, Cryosphere and Applications communities including cloud/snow mask over land, high spatial resolution (1km) aerosol optical depth and type, surface bidirectional reflectance factors (BRF) and BRDF, and snow grain size and sub-pixel snow fraction for the snow-covered regions.

The goal of this paper is to give a systematic description of MAIAC Collection 6 algorithm and its products, along with current limitations, provide quality assurance discussion and recommendations for use. For practical reasons, here we focus on MAIAC cloud detection, aerosol retrieval and atmospheric correction over land; snow detection, over water processing, and smoke plume height retrieval will be described elsewhere. This paper is structured as follows: an overview of MAIAC processing is given in section 2; section 3 describes MAIAC radiative transfer model and look-up tables followed by

MAIAC processing components, including cloud/snow detection and aerosol type selection (sec. 4), determination of SRC (sec. 5) and aerosol retrieval procedure (sec. 6-7); shadow detection (sec. 8) and atmospheric correction (sec. 9). The last section 10 describes MAIAC (MCD19) product and quality assurance (QA) specification.

## 2. Overview of MAIAC Processing

The block-diagram of MAIAC processing over land, which implements a sliding window algorithm, is shown in Figure 1:

1) The received L1B data are gridded (Wolfe et al., 1998), split in 1200 km tiles, and placed in a Queue with the previous data. We are using the area-weighted gridding method which achieves better agreement with the ground tower data over heterogeneous surfaces as compared to the nearest neighbor resampling method (Zhang et al., 2014). The 1km MODIS bands are gridded to 1km resolution, the 500m bands (B1-B7) are gridded to 1km and 500m, and 250m bands (B1-B2) are gridded to 250m. The 500m and 250m bands are nested in the 1km grid. For convenience, here is the list of MODIS bands

B1-B12 where MAIAC reports BRF at 1km (0.645 (B1), 0.856 (B2), 0.465 (B3), 0.554 (B4), 1.242 (B5), 1.629 (B6), 2.113 (B7), 0.412 (B8), 0.442 (B9), 0.487 (B10), 0.530 (B11), 0.547 (B12) ).

2) MAIAC uses different spatial scales for processing, e.g. 1km grid cells (or pixels), 25km blocks, and 150km mesoscale areas. Specialized C++ classes and structures handle processing in different time–space scales. The Queue (Q) (Lyapustin et al., 2012a) holds between 4 (at the poles) and 16 (at the equator) days of imagery. For every observations, MODIS data are

stored as layers for the required bands along with the retrieval results. A dedicated Q-memory accumulates ancillary information for each 1km grid cell for cloud/snow detection, aerosol retrieval and AC. Q-memory stores the following data at 1km resolution: the reference clear-sky image for bands B1, B3, B7 (see Lyapustin et al., 2008), spectral BRDF for bands B1-B8; BRDF-normalized (to nadir view and solar zenith angle $SZA=45°$) bidirectional reflectance factors $BRF_n$ for bands B1,B2,B7; 2×2 standard deviation of 500m pixels for B1, B3; normalized difference vegetation index (NDVI); 11μm brightness temperature ($Tb_{11}$, band 31); 4μm (B22) - 11μm (B31) ($Tb_{4-11} = Tb_4 - Tb_{11}$) and 11μm - 12μm (B32) ($Tb_{11-12} = Tb_{11} - Tb_{12}$) spectral thermal contrasts. Below, we will use notation q.$Tb_{11}$ for the Queue brightness temperature, as an example.



Prior to processing, we compute covariance of the latest measurements with the reference clear-sky image (Lyapustin et al., 2008) in bands B3, B1 and B5 for $25 \times 25 \text{km}^2$ blocks. When covariance is high, indicating high probability for the 'confidently clear conditions', we set the corresponding flag q.iFlag_HighCov=1 used in the cloud detection later.

2) The column water vapor is computed for the last tile using MODIS near-IR channels B17-B19 located in the water vapor absorption band 0.94 μm (Lyapustin et al., 2014). This algorithm is a modified version of Gao and Kaufman (2003). It is fast, unbiased and has the average accuracy of ±(5-15)% over the land surface (ATBD; Martins et al., 2017) whereas standard NIR CWV product (MOD05) has a known wet bias of 5-20% (e.g. Albert et al., 2005; Prasad and Singh, 2009; Liu et al., 2013).

Column water vapor is retrieved over cloud-free land pixels and over clouds. In the latter case, it represents water vapor above the cloud.

3) The cloud mask box includes dynamic land-water-snow classification, determination of aerosol type (background vs smoke or dust) and cloud detection. The smoke/dust test is based on the enhanced shortwave absorption and effective particle size (Lyapustin et al., 2012c) and requires knowledge of spectral surface BRDF. At 1km resolution, brightness and spatial contrasts of smoke plumes can be high, leading to a competition between smoke and cloud detection. With optimal combination of different cloud tests and smoke detection, MAIAC provides aerosol retrievals for most of smoke plumes with minimal cloud leak.

4) When snow is detected, MAIAC derives surface reflectance using regional climatology AOD and computes snow grain size (SGS) and sub-pixel snow fraction (SF) as a best fit to obtained spectral reflectance. The reflectance is modeled as a linear mixture of snow BRDF and land BRDF, which is stored in the Queue from retrievals prior to snow detection. Snow reflectance is modeled with semi-analytical model (Kokhanovsky and Zege, 2004) with consideration for surface roughness (Lyapustin et al., 2010). Assuming a background soot concentration, the snow reflectance depends only on SGS, which is computed from measured spectrum along with SF. The algorithm is analytical, fast and robust due to a large difference between spectra of regular land types and snow. The derived SGS may have a large uncertainty because of rapid metamorphism of aging snow and pollutions from the atmosphere which reduce snow reflectance. We validated SGS retrievals over pure snow (Lyapustin et al., 2009). Algorithm description and validation of snow fraction using high resolution Landsat data will be given elsewhere.

5-6) Using knowledge of spectral BRDF and SRC at 1km, MAIAC retrieves AOD at 1km resolution. The following post-processing uses several filters to detect residual clouds and smooth the noise from gridding uncertainties. This step significantly increases quality of AOD and of atmospheric correction.

7) The combination of MAIAC cloud mask and AOD filters 6) detects majority of clouds. The next step 7) detects cloud shadows using geometric approach and our knowledge of BRDF.

8) Determination of spectral regression coefficient (SRC) required for aerosol retrieval.

9) In cloud-free and clean to moderately hazy ($AOD_{0.47} < 1.5$) conditions, MAIAC atmospheric correction computes spectral BRF at 1km resolution (bands B1-B12), and at 500m resolution (bands B1-B7). Combining BRF from the latest observation





with previous values stored in the Queue, MAIAC performs BRDF inversion in 1km bands 1-8 providing 3 parameters of the Ross-Thick Li-Sparse (RTLS) BRDF model (Lucht et al., 2000).

10) Evaluate aerosol layer height for detected smoke pixels using thermal technique (Lyapustin et al., to be submitted).

11) At the end of processing, the Q-memory is updated for cloud-free pixels under clean atmospheric conditions.

**3. Radiative Transfer Model and Look-up Tables**

MAIAC radiative transfer (RT) model uses a semi-empirical Ross-Thick Li-Sparse (RTLS) BRDF model (Lucht et al., 2000) used in the operational MODIS BRDF/albedo (MOD35) algorithm (Schaaf et al., 2002). This is a linear model, represented as a sum of Lambertian, geometric-optical, and volume scattering components:

$$\rho(\mu_0, \mu, \varphi) = k^L + k^G f_G(\mu_0, \mu, \varphi) + k^V f_V(\mu_0, \mu, \varphi).$$ (1)

It uses predefined geometric functions (kernels) $f_G$, $f_V$ to describe different shapes as a function of view geometry ($\mu_0, \mu, \varphi$ - cosines of solar and view zenith angles, and relative azimuth). The kernels are independent of the land conditions. The BRDF of a pixel is characterized by a combination of three kernel weights, $\vec{K} = \{k^L, k^G, k^V\}^T$.

MAIAC RT model is based on the semi-analytical Green's function (GF) solution for the TOA reflectance (Lyapustin and Knyazikhin, 2001). When combined with linear RTLS model, the GF solution provides an explicit expression for the TOA

reflectance as a function of the RTLS model parameters (Lyapustin et al., 2011a):

$$R(\mu_0, \mu, \varphi) = R^A(\mu_0, \mu, \varphi) + k^L F^L(\mu_0, \mu) + k^G F^G(\mu_0, \mu, \varphi) + k^V F^V(\mu_0, \mu, \varphi) + R^{nl}(\mu_0, \mu).$$ (2)

Here, $R^A$ is atmospheric path reflectance. Functions $F^L$, $F^V$, $F^G$, $R^{nl}$ depend on view geometry and aerosol properties. They are also weakly non-linear functions of $\vec{K}$-parameters which accounts for multiple reflections of sunlight between the land surface and the atmosphere. This dependence is analytical and is conveniently handled by the second iteration during the

atmospheric correction. The details for computing functions $F^j$, $R^{nl}$ are given by Eqs. (1-25) in *Lyapustin et al.* (2011a). In brief, they are expressed via eight basic functions which represent different hemispheric integrals from downward path radiance, atmospheric Green's function (or bidirectional upward diffuse transmittance), and RTLS kernels (1, $f_V$, $f_G$). These functions, along with path reflectance, are pre-computed and stored in the look-up table (LUT). Throughout this paper, AOD refers to the Blue wavelength ($AOD_{0.47}$).

Eq. 2 is used in MAIAC cloud detection, selection of aerosol type, and in the atmospheric correction. For SRC and aerosol retrieval, we also use Lambertian Equivalent Reflector (LER) approximation,

$$R(\mu_0, \mu, \varphi) \cong R^A(\mu_0, \mu, \varphi) + \rho(\mu_0, \mu, \varphi)T^d(\mu_0)T^u(\mu)/(1 - s\rho(\mu_0, \mu, \varphi)),$$ (3)



where $T$ is the total downward (d) and upward (u) transmittance, and $s$ is spherical albedo of atmosphere. Over the water, equation (3) is modified to account for the diffuse reflectance $\rho^w$ of underlight, representing water leaving radiance:

$$R(\mu_0,\mu,\varphi) \cong R^{A+s}(\mu_0,\mu,\varphi) + \rho^w(\mu_0,\mu,\varphi)T^d(\mu_0)T^u(\mu) , \qquad (4)$$

where $R^{A+s}$ contains Fresnel reflectance from wind-ruffled ocean surface and a whitecup contribution (Koepke, 1984) in
addition to the atmospheric path reflectance.

Since (Lyapustin et al., 2011a), MAIAC aerosol models and LUTs were simplified considerably. We abandoned approach of mixing fine and coarse aerosol fractions in favor of using regional aerosol models based on AERONET (Holben et al., 1998) climatology, e.g. (Dubovik et al., 2002; Eck et al., 2010). C6 MAIAC uses eight regional aerosol models and respective LUTs (see sec. 6) over land, including a separate dust model. Since MAIAC processing is tile-based and inherently regional,
it only reads the required regional LUT or LUTs without overloading operational memory. This allows us to discretize the world map in sufficient detail to account for the regional aerosol variability.

Each LUT is computed with full multiple scattering: all functions are first computed using LUT-generation software based on scalar code SHARM (Lyapustin, 2005), and the atmospheric path reflectance is then replaced with vector solution from code IPOL. The discrete ordinates code IPOL was recognized as the best overall among 10 different vector codes which
participated in the recent inter-comparison study (Emde et al., 2015).

Each LUT is generated for the standard P=1 and reduced (P=0.7) pressure levels (normalized to the standard pressure 1013.25mB) in order to account for surface height variations using linear interpolation. Computations with P=0.7 are done for wavelengths shorter than 0.66μm.

As before, the spectral gaseous absorption used in the LUT radiative transfer was obtained based on the line-by-line
calculations (Lyapustin, 2003) for MODIS spectral response functions. The computations include absorption of 6 major atmospheric gases ($H_2O$, $CO_2$, $CH_4$, $NO_2$, $CO$, $N_2O$) calculated for the HITRAN-2008 (Rothman et al., 2009) database using the Voigt vertical profile, and the Atmospheric Environmental Research (AER) continuum absorption model (Clough et al., 2005). The LUT is generated for a fixed column water vapor, $W_0$=0.5 cm. In MODIS Red band where WV absorption is maximal, the atmospheric path reflectance is also generated for WV=6cm, and linear interpolation is used to account for the
WV variations.

For the pressure and WV-correction, the surface-reflected signal is multiplied by the two-way direct transmittance of the well mixed gases and water vapor $t^g(P)t(W_0,W)$:

$$t^g(P) = \exp(-(1-P)\tau^g m) , \; m = \left|\mu\right|^{-1} + \mu_0^{-1} , \qquad (5)$$

$$t(\lambda,W) = \exp(-a(\lambda,W)m^{b(\lambda,W)}) , \; t(W_0,W) = t(\lambda,W)/t(\lambda,W_0) . \qquad (6)$$



Above, $m$ is an atmospheric air mass, and parameters $a$ and $b$ are obtained by fitting the angular dependence of the water vapor in-band transmittance. Expression (5) is a modified form of equation for the broad-band transmittance of water vapor (Schmid et al., 2001).

Finally, LUTs are computed for a relatively sparse angular grid ($\Delta\mu_0 = \Delta\mu$ =0.05 for the range $\mu$=0.4 − 1 (0° - 66.42°),

$\mu_0$=0.15–1 (0° - 81.37°) and $\Delta\varphi$= 9°) and 12 AOD values, {0.05, 0.1, 0.2, 0.3, 0.4, 0.55, 0.75, 1., 1.4, 2.0, 2.8, 4.0} giving the size of 45.7MB per a regional aerosol model. Rayleigh LUT (AOD=0) is generated separately.

Ordinarily, generating LUT-based TOA reflectance for shortwave channels requires two 3D-interpolations in angles at P=1, 0.7, with following linear interpolation in pressure for a number of required functions per pixel. To optimize MAIAC processing, we introduced an intermediate-scale radiative transfer RT-containers for 5km boxes. Each box is characterized

by an average view geometry, mean water vapor and surface pressure, representing the average height. For each box, we compute the required functions for 13 AOD LUT nodes. After that, specific MAIAC processing for any 1km pixel within a given 5km box only requires an additional linear interpolation in AOD using functions from the RT-container, and an analytical WV-correction for the surface-reflected signal. The 5km RT-containers ($RT_5$) are generated for boxes with cloud-free pixels, and are stored as a layer in the Queue. This approach allows us to use the same RT-container repeatedly at

different stages of MAIAC processing which reduces computational cost by at least a factor of 25. While theoretically such approach may create biases at short wavelengths (B3 (0.47µm) and B8 (0.412µm)) on the boundaries of boxes with sharp height gradient, a very extensive near-global testing did not reveal any noticeable difference in AOD or surface BRF compared to the accurate 1km pixel-level interpolation in view geometry and pressure.

Prior to processing of a new MODIS observation, we compute a spectral deviation of measurements ($M$) from the expected

theoretical ($T$) clear-sky (AOD=0) TOA reflectance,

$$\delta_\lambda = R_\lambda^M - R_\lambda^T(\tau^a = 0).$$ (7)

$\delta_\lambda$ is computed for 5 bands (B1, B3, B8, B5, B7), and is indicative of atmospheric perturbations from clouds and aerosols as illustrated in Fig.2 for band B3. Because MAIAC freezes BRDF retrievals when snow is detected such that the Queue BRDF always represents snow-free land reflectance, $\delta_\lambda$ also contains spectral signatures of snow and is used in the snow detection.

Using an estimate of Jacobian in the Blue band (B3), $\partial R/\partial\tau^a \approx (R^T(0.05) - R^T(0))/0.05$, it is easy to obtain an initial assessment of AOD,

$$\tau_0^a = \delta_{0.47}(\partial R/\partial\tau^a)^{-1},$$ (8)

which appears quite accurate except over bright surfaces. To guide aerosol retrievals, we also evaluate a theoretical uncertainty of AOD in response to uncertainty in the surface reflectance at 0.47µm which is assumed as δρ=$max${0.002;

0.04$RTLS(\mu_0,\mu,\varphi)$}. This estimate is based on the extensive evaluation of MAIAC retrieval accuracy, although it may



overestimate the uncertainty over bright surfaces. Given δρ and neglecting other contributions, e.g. from variation in the aerosol model, the AOD uncertainty is:

$$\delta\tau^a = \delta R(\partial R / \partial \tau^a)^{-1},\tag{9}$$

where $\delta R = R^T(0; RTLS + \delta\rho) - R^T(0; RTLS)$ is computed for the perturbed BRDF. As the surface becomes brighter,

the sensitivity of measurements to aerosol $(\partial R / \partial \tau^a)$ decreases and AOD uncertainty (9) grows. This uncertainty is used in MAIAC as a measure of the surface brightness guiding the aerosol retrieval algorithm.

For optimization, $RT_5$ container is initially filled everywhere for AOD=0, 0.05 in order to compute deviation from the clear-sky (7) and evaluate initial AOD and its uncertainty (8-9). The rest of $RT_5$ container (AOD≥0.1) is filled only for the boxes containing cloud-free pixels after detection of *Reliable Clouds* (tests C1-C4).

**4. Cloud Mask**

The cloud mask box in Fig.1 consists of dynamic Land-Water-Snow (LWS)-classification, and cloud mask tests combined with the aerosol type selection. MAIAC uses both local (pixel-level) and contextual information from the surrounding area. The latter comes from the 150km mesoscale boxes where we evaluate *min* and *max* values of brightness temperature ($Tb_{11}$), reflectance in MODIS cirrus band (*B26*) $r_{1.38}$, column water vapor, number of (internally) detected fire hot spots, and the

number of previously detected snow pixels based on the Q-information. This non-local information appears very useful, for instance, for choosing more or less conservative pixel-level snow or smoke detection algorithm etc.

MAIAC needs to know the state of the surface (land/water/snow/ice) to select the proper processing path. For this purpose we developed the dynamic Land-Water-Snow classification (LWSC) from daily observations. It uses several tests and a decision tree. The LWSC logic and details of snow processing will be described separately.

The conventional cloud mask algorithms, e.g. (Ackerman et al., 1998; 2006), makes cloud detection and classification based on groups of tests identifying cloud types. As MAIAC does not require cloud typing, its tests are applied sequentially, and processing terminates as cloud is detected. MAIAC cloud mask algorithm is only a beginning of cloud detection, which is consecutively enhanced by filters following aerosol retrieval and then by the atmospheric correction component of MAIAC.

**4.1 Reliable Clouds**

The first group of tests, which have low interference with the smoke/dust detection, includes the bright, cold/high, and spatial variability tests:

1)   *Bright cloud test*: Measured reflectance exceeds theoretical value at maximal LUT AOD=4 with a certain threshold:

$R^M > R^T_{max} + thresh$, where *thresh*=0.1 for the Sahara region and 0.03 otherwise.        (C.1)



The test uses the shortest MODIS channel B8 (0.412μm), where reduction of TOA reflectance by absorbing aerosols (smoke/dust) and the difference in reflectance with non-absorbing clouds is maximal.

2)      *Cold (high) cloud test*: Measured brightness temperature $Tb_{11}$ is lower by 30° or more than the expected value for this pixel (either q.$Tb$ or a maximal meso-scale value $Tb_{max}^{Meso}$), combined with high "cirrus" band reflectance or high thermal contrast $dTb_{4-11}$:

$$Tb_{11} < 283 \;\; \text{AND} \;\; Tb_{11}+30 < min(\text{q}.Tb, \; Tb_{max}^{Meso}) \;\; \text{AND} \;\; ( R_{1.38}^M > 0.03 \;\; \text{OR} \;\; dTb_{4-11} > 10). \tag{C.2}$$

The 30° difference corresponds to an altitude difference of ~4.5km for an average lapse rate of 6.5° km$^{-1}$.

*3)   High cloud test (for pixels with elevation below 2.5km):*

$$H < 2.5\text{km} \;\; \text{AND} \;\; R_{1.38}^M > 0.035 \;\; \text{AND} \;\; dTb_{4-11} - \text{q}.dTb_{4-11} > 5. \tag{C.3}$$

*4) Spatial variability test* (Lyapustin et al., 2012c): 2×2 standard deviation of 500m pixels nested in 1km grid cell significantly exceeds the clear-sky threshold (q.σ) stored in the Queue:

$$\sigma > \text{q}.\sigma\sqrt{\mu} + thresh, \text{ where } thresh=(\text{q}.\sigma)^{max}. \tag{C.4}$$

Here, the multiplier $\sqrt{\mu}$ approximately accounts for the pixel growth and higher overlap between scan lines with scan angle, and the resulting reduction of contrast. The threshold *thresh* depends on the surface variability and represents maximal contrast over a given pixel and its nearest neighbors. If a fire hot-spot is detected in the mesoscale range of given pixel, the *thresh* is increased by a factor of 2-3.5 depending on pixel's proximity to the hot-spot. Also, the *thresh* is increased in 'confidently clear conditions' (q.iFlag_HighCov=1) by a factor of 2 to avoid false cloud detection over high contrast, e.g. urban, land surfaces.

Test (C.4) is applied globally over land using MODIS Red band B1. It works well over darker soils and vegetated surfaces, and is successful capturing many small popcorn cumulus clouds which is a major issue and source of errors in the remote sensing. Over deserts, the surface is bright in the Red band, and the contrast with clouds is significantly reduced. In these cases selected as *q.NDVI*<0.2, test (C.4) is repeated for the Blue (B3) band using the fixed threshold *thresh*=0.012.

Finally, to "clean" the cloud boundaries where the contrast is often reduced due to lower sub-pixel cloud fraction, we repeat the above procedure using the reduced threshold (0.6 *thresh*). This second iteration is applied to pixels neighboring detected clouds.



## 4.2 Smoke/Dust Detection

The smoke test described in (Lyapustin et al., 2012b-c), uses MODIS Red, Blue and Deep Blue (DB) bands B1 (0.646 μm), B3 (0.47 μm) and B8 (0.412 μm). The developed test 1) isolates atmospheric aerosol reflectance, and 2) compares the measured reflectance at shortest wavelength (0.412 μm) with that predicted from the Red-Blue region using the background

aerosol model. For absorbing aerosol containing both black and brown carbon, the measured aerosol reflectance at 0.412 μm is lower than predicted due to both 1) more absorption caused by more multiple scattering at 0.412 μm, and 2) increased shortwave absorption (by brown carbon for smoke and by iron compounds for dust) from increasing imaginary refractive index at 0.412 μm as compared to the Red-Blue region.

The smoke test first computes an aerosol reflectance in the Red, Blue and DB channels by subtracting the Rayleigh (path)

reflectance and the full surface-reflected signal at TOA from the measurement:

$$R_\lambda^{Aer} = R_\lambda^{Meas} - R_\lambda^{Molec} - R_\lambda^{Surf}(\tau^a).$$ (S.1)

The last term is computed using $\tau_0^a$ (Eq. 8) evaluated with the background aerosol model and known spectral surface BRDF. Assuming a power law spectral dependence, $R_\lambda^{Aer} \sim \lambda^{-b}$, we compute the equivalent Angstrom exponent $b$, or the size parameter (SP) using the Red and Blue channels,

$$SP = R_{0.646}^{Aer} / R_{0.466}^{Aer},$$ (S.2)

and the absorption parameter (AP) as a ratio of measured and predicted "aerosol reflectance",

$$AP = R_{0.412}^{Aer,Meas} / R_{0.412}^{Aer,Pred}, \text{ where } R_{0.412}^{Aer,Pred} = R_{0.466}^{Aer}(\frac{0.466}{0.412})^b.$$ (S.3)

The idea behind this test is similar to the OMI Aerosol Index (AI) detection (Torres et al., 1998; 2007): to the first order approximation, the clouds, which have spectrally neutral behavior, or non-absorbing aerosols, would give the AP values

close to unity, whereas the absorbing aerosols would result in the lower AP values. Theoretical simulations (Lyapustin et al., 2012b-c) show a robust aerosol-cloud separation at $AOD_{0.47}>0.5$ based on AP-SP indices.

As specific aerosol absorption is a function of many parameters including type of the burning material and smoldering to flaming fraction ratio for smoke, or mineral composition including hematite content for mineral dust, we first define the approximate parameterized cloud properties based on theoretical simulations:

$AP_{Cloud} \cong 0.97 - 0.06(2 - \mu - \mu_0)$ and $SP_{Cloud} \cong 1.15 + 0.15(2 - \mu - \mu_0)$. (S.4)

Then, the smoke/dust tests are implemented based on "separation from the clouds" as follows:

If $AP_{ij} < AP_{Cloud} - 0.03$ AND $SP_{ij} < SP_{Cloud}$ AND $dTb_{4-11} - q.dTb_{4-11} < TH_S \rightarrow$ Smoke; (S.5)





If $AP_{ij} < AP_{Cloud} - 0.03$ AND $dTb_{4-11} - q.dTb_{4-11} > TH_D \rightarrow$ Dust. (S.6)

As smoke generally does not exhibit thermal contrast, the thermal threshold is low, $TH_S=1.5K$. This is not true near the fire hot spots: based on extensive analysis of MODIS data, we parameterized the threshold in this case as a function of AOD, $TH_S=2.5+0.5AOD$.

To detect most dust for the Sahara region where dust is the dominant aerosol type, the threshold is set to be low $TH_D=1.5$; for other dust regions, the threshold is increased to $TH_D=3$.

The dust test (S.6) often misclassifies thin cloud edges as dust. For this reason, we avoid the 2-pixel zone adjacent to the detected clouds, and limit the dust test to the dust regions only (see sec. 6.1).

## 4.3 Final Cloud Mask

The final cloud test combines analysis of "cirrus" band reflectance $R^M_{1.38}$ (B26) and thermal contrast $dTb_{4-11}$. This test has evolved during several years of development. Initially, we followed MODIS cloud detection (Ackerman et al., 2006) and used the "cirrus" test alone. Contrary to MODIS though, which uses a single global threshold $R^M_{1.38} > 0.035$ except in winter and at high elevations, we set a dynamic threshold as a function of the retrieved column water vapor. This way, we could decrease the cloud detection threshold down to 0.008, still well above the noise level in band 26, and detect either very thin

cirrus or lower clouds with partial absorption by water vapor above the cloud. Figure 2 gives an illustration of the "cirrus" band reflectance showing both high and weak but spatially coherent signal from lower clouds, which may not be easily detectable in the RGB bands.

Figure 2 also shows the atmospheric thermal contrast $dTb^A_{4-11}=dTb_{4-11} - q.dTb_{4-11}$. *Ackerman et al.* (2006) mentions high information content of $dTb_{4-11}$ for cloud detection, but also states that it is hard to use globally due to its significant

variability from the land surface. By characterizing surface component $q.dTb_{4-11}$ on clear days, MAIAC can separate an atmospheric variation $dTb^A_{4-11}$ which significantly raises information content of this spectral thermal signature for the cloud detection. Analysis of near-global MODIS data showed that $R^M_{1.38}$ and $dTb^A_{4-11}$ usually carry similar information for cloud detection, but sometimes it is complementary to that of the "cirrus" channel (see Fig. 2), so the joint test gives a better cloud detection.

The C6 MAIAC $R^M_{1.38}$ - $dTb^A_{4-11}$ test works as follows.

    1) Detect clouds with high $dTb^A_{4-11}$

$dTb^A_{4-11} > thresh$, where $thresh=10$. (C.5)




The threshold is increased to 14 in two cases: 1) A snow-covered surface usually has a very low thermal contrast ($q.dTb_{4\text{-}11}$). When snow melts, an exposed bare soil may exhibit a much higher contrast; thus, the threshold increase helps avoid a commission error of cloud detection. Snow ablation is identified when $R_{2.13}^{M} > R_{0.64}^{M}$ and snow has been detected previously for a given pixel, but was not detected currently; 2) under 'confidently clear conditions' (q.iFlag_HighCov=1).

2)   Detect clouds with high product:

$$dTb_{4-11}^{A}(R_{1.38}^{M}/0.005) > thresh, \tag{C.6}$$

where 0.005 is close to the noise level of B26, and *thresh* is set as 25 for the Sahara region, 15 for bright surfaces ($R_{2.13}^{M} > 0.3$ OR $q.dTb_{4\text{-}11}>5$), and 6 otherwise. This test is designed for conditions when neither the "cirrus" band reflectance nor the thermal contrast are high enough to reliably detect clouds, but their product can do it.

3)   11-12µm difference $dTb_{11-12}^{A}$ test ($dTb_{11-12}^{A}=dTb_{11\text{-}12} - q.dTb_{11\text{-}12}$). This test is only applied within 2-pixels on the border of detected clouds. According to *Ackerman et al*. (2006), the $dTb_{11\text{-}12}$ difference is positive and increases for clouds and decreases for dust, although not universally. The $dTb_{11-12}^{A}$ test is set as follows:

$$dTb_{11-12}^{A}>0.5 \text{ AND } dTb_{4-11}^{A}>2. \tag{C.7}$$

This concludes the tests within the cloud mask block. The following test is applied during the atmospheric correction
routine:

     4)   Over Dark Dense Vegetation (DDV) defined as q.NDVI>0.75, the low B1 reflectance of the surface, often associated with a good degree of homogeneity at 1km scale, allows for an enhanced sub-pixel cloud detection during stable surface conditions. This filter consists of two tests: a) comparison of geometrically-normalized B1 reflectance with the Queue-value at 1km:

$$BRF_{n,B1} / q.BRF_{n,B1} > 1.35. \tag{C.8}$$

Over dense vegetation with the Red band reflectance as low as 0.02-0.03, this test can detect clouds with reflectance difference of ~0.007-0.01.

Over homogeneous 1km DDV pixels defined as $q.\sigma_{B1} < 0.006$, the nested 500m pixels should have a similar reflectance to that of the 1km grid cell. The second test *b*) checks measured sub-grid variability and detects sub-pixel clouds based on high
ratio of 500m BRF in B1 to the 1km value computed from the RTLS model,

   $$\rho_{500,B1} / RTLS_{B1} > 1.8. \tag{C.9}$$

The thresholds in tests 8-9 were selected based on extensive processing of MODIS data and have a low commission error.



## 5. Spectral Regression Coefficient

Retrieval of spectral regression coefficient (SRC, box 8), or spectral SR ratios $b_{37}$=B3/B7 and $b_{34}$=B3/B4, is a central component of MAIAC required for aerosol retrievals. It runs independently and provides separation between atmospheric and surface contributions.

The C6 MAIAC SRC retrieval has changed completely. The early version (Lyapustin et al., 2011b) used a multi-day minimization for all cloud-free pixels in the $25 \times 25 \text{km}^2$ area. While this approach was successful overall, it could generate an occasional random SRC-bias for the whole block creating AOD "blockiness" at 25km scale which further propagated into the surface reflectance. To resolve this instability, we developed a new pixel-based approach which is much simpler and gives more accurate AOD. The new approach uses the minimum reflectance method: SRC ($b_{37}$) is found as a minimal ratio

of surface reflectance, e.g. $\rho^*_{0.47}/\rho^*_{2.13}$, over the two months period. For each observation, an apparent LER $\rho^*$ is computed from TOA measurements using Eq. 3 assuming some regional background aerosol level, e.g. $AOD_{0.47} \sim 0.05$. As the uncompensated aerosol increases $\rho^*_{0.47}$ in the Blue band where most surfaces are "dark", selection of the minimal value over time provides a reliable SRC estimate. This technique is cloud-resistant as residual clouds increase the ratio $\rho^*_{0.47}/\rho^*_{2.13}$; however, it is sensitive to undetected shadows, and therefore SRC retrieval is preceded by the shadow detection (box 7, Fig.

15  1).

While the minimum reflectance method is a powerful generic technique, it should be used with caution. For instance, the described algorithm can only reduce SRC over time; it is also prone to accumulating erroneous very low values. In reality, seasonal surface change and annual variation of sun zenith angle create both upward and downward patterns. As one of the measures addressing these issues, MAIAC uses two independent lines of SRC retrieval ($b_1$ and $b_2$, Fig. 3) starting on odd and

even months, and each taking two months to re-initialize. This way SRC is updated monthly and can both decrease and increase over time. The two-month initialization period was selected for the MODIS observation frequency empirically to account for possible periods of high cloudiness and/or high aerosol concentration. Under favorable conditions, the SRC is updated as soon as the new minimum is found, along with the update of both lines $b_1$ and $b_2$. This way, the SRC used in aerosol retrievals can be updated more frequently than once per month with the new low value, and once a month in case of

increasing SRC trend.

The land surface is considerably brighter at $2.13\,\mu\text{m}$ compared to the Blue wavelength. This results in spectral dependence of the BRDF shape and in SRC dependence on the view geometry. To account for that, MAIAC SRC is computed for 3 angular bins carefully selected to optimize aerosol retrievals over bright deserts where the AOD error sensitivity is maximal. Current bins represent forward scattering ($\varphi \leq 90°$), backscattering ($\mu < 0.95$, $\varphi > 90°$), and "nadir" direction ($0.95 < \mu \leq 1$, $\varphi > 90°$), the

latter introduced to represent regions of the land hot spot for tropics/sub-tropics and near-nadir views when the sun is near zenith. A linear interpolation between bins is used within $\Delta\mu \leq \pm 0.01$ across bin-boundaries.

Before using the LER model, we studied the full radiative transfer with anisotropic surface model where SRC is used to predict the Blue band BRDF from the BRDF at 2.13 µm. That approach was computationally more expensive and still





required angular binning of SRC. Besides, we found that it was also sensitive to the B7 BRDF errors over bright surfaces occasionally producing AOD outliers. Over bright surfaces, small errors in the BRDF shape at 2.1 μm can result in relatively large errors in the surface-reflected diffuse radiance because of high values of the BRDF shape parameters ($k_v$, $k_g$). The BRDF errors can arise from uncertainties of gridding, limitations of the RTLS model (not exactly matching the real

distribution), or rarely, unstable RTLS inversions. Using only TOA measurements at 2 wavelengths, the LER approach eliminates the BRDF model-based sources of uncertainty and provides more stable AOD retrieval with better AERONET comparison. The current C6 MAIAC uses LER surface model for both SRC and aerosol retrievals.

## 6. Aerosol Retrievals

### 6.1 Aerosol Models

The geographic distribution of regional background aerosol models over land used in MAIAC processing is shown in Figure 4. MAIAC uses 8 different models listed in Table 1. Model properties are given in terms of volumetric size distribution (e.g., Dubovik and King, 2000) with radius ($R_v$) and standard deviation ($\sigma_v$) for the fine and coarse modes, their ratio of concentrations ($C_v^C / C_v^F$), real ($m$) and imaginary ($k$) refractive index, absorption Angstrom Exponent (AAE) defined with respect to spectral dependence of $k$, and spherical (Mie) aerosol fraction. The imaginary refractive index is assumed to be

spectrally dependent at $\lambda < \lambda_0 = 0.66$ μm, $k(\lambda) = k(\lambda_0)(\lambda/\lambda_0)^{-AAE}$, and constant for longer wavelengths. The aerosol models can be either static with parameters fixed, typical of arid environment, or dynamic (Remer and Kaufman, 1998) with parameters depending on AOD. Growth of volumetric radius with AOD represents hygroscopic growth of aerosol particles associated with AOD increase. It is typical for regions with moderate-to-high humidity. Model parameters (size distribution, ratio of volumetric concentrations, refractive index) are generally representative of the AERONET regional climatology (e.g.,

Dubovik et al., 2002) with empirical adjustments aimed at achieving a better match of retrieved AOD to AERONET sunphotometer data.

Dynamic model 1 based on the GSFC AERONET site represents east coast USA with high summertime humidity. More arid climate of the western USA is represented by Model 2, with some contribution of dust particles and larger coarse fraction. Model 3 with high absorption was developed to model polluted environment of Mexico City. European model 4 has a higher

absorption, but otherwise is the same as the East Coast USA model 1. Model 5 representing "industrial world" China was developed based on Beijing AERONET model with an adjustment for absorption. "India" model 8 is similar to 5 but with higher absorption coming from agricultural biomass burning (seasonal), cooking and transportation (e.g., Singh et al, 2017). The biomass burning "cerrado" model 7 of sub-equatorial Africa was developed based on AERONET Mongu site. Finally, the desert dust model 6 was based on Dubovik et al. (2002) climatological model for the Solar Village site.

The transparent yellow shape in Fig. 4 maps the world region where the dust test is conducted and AOD is retrieved with the background or the dust model depending on the dust test outcome. In MAIAC C6 version, we still use the regional



background model for aerosol retrieval and atmospheric correction even if "Smoke" was detected. The next version will use a joint AOD-SSA (single scattering albedo) retrieval algorithm for areas with detected Smoke. This algorithm has already been developed and is in the testing/tuning phase.

Lack of seasonal dependence of aerosol models and LUTs is one of MAIAC C6 limitations. It does not account for regional aerosol seasonality, for instance periods of biomass burning and variations in humidity. As a result, current AOD product may show seasonal biases, for instance over India during post-monsoon biomass burning (Mhawish et al., in review). This issue will be fixed in the next version of MAIAC.

## 6.2 Aerosol Algorithm

The aerosol algorithm depends on the brightness of surface which is characterized using the uncertainty parameter $\delta\tau^a$. Over dark surfaces ($0 \leq \delta\tau^a < 0.05$), the AOD retrieval routine, first, evaluates LER in B3 (0.47 µm):

$$\rho_{0.47} = b_{37}\rho_{2.13}, \tag{15}$$

and then computes AOD by matching the LUT-based theoretical reflectance to the measurement,

$$R_{0.47}^T(\tau^a) = R_{0.47}^M. \tag{16}$$

The LER $\rho_{2.13}$ is obtained by atmospheric correction from the measurement $R_{2.13}^M$ with current AOD used in the aerosol retrieval loop. However, when Smoke/Dust is detected, or LER $\rho_{2.13}$ is significantly different from the BRDF model value, $\rho_{2.13} < 0.5 RTLS_{2.13}$ or $\rho_{2.13} > 2 RTLS_{2.13}$ ($\rho_{2.13} > 1.5 RTLS_{2.13}$ for bright surfaces when $k_{2.13}^L > 0.25$) which usually indicates undetected clouds or cloud shadows, we use the BRDF model as LER, $\rho_{2.13} = RTLS_{2.13}$.

The Dark Target algorithm (Levy et al., 2013) is prone to overestimating AOD as surface brightness increases. A typical example of high bias in the VIIRS aerosol algorithm is given by Fig. 7d from (Superczinsky et al., 2017). While MAIAC implementation is different from the VIIRS (Jackson et al., 2015), it faces the same general issue. Over brighter surfaces, as sensitivity of measurements to AOD decreases, the effect of the surface-related errors increases. The latter span the range of errors from gridding to errors from the lagged SRC-characterization with the time series method. Over bright barren land, the latter source, essentially related to the change in the average sun angle during the 2-month lag period, becomes more important over mountainous regions with terrain slope variations. Statistically, most surface-related errors, including those from gridding, should be symmetric about zero. However, because we do not accept negative AOD, the net effect is a positive bias.

One more error source is characterization of the angular dependence of SRC. As surface brightness increases, the difference in the BRDF shape between darker 0.47 and much brighter 2.13 µm channels (angular dependence of SRC) increases. To



reduce this effect, we added minimization of the Blue/Green band ratio where the surface brightness and BRDF shapes are much closer. The resulting AOD retrieval is based on minimization of the following function:

$$F(\tau^a) = w_1(1 - R_{0.47}^T(\tau^a)/R_{0.47}^M)^2 + w_2(1 - [\rho_{0.47}(\tau^a)/\rho_{0.55}(\tau^a)]/b_{34})^2, \qquad (17)$$

where the weights of B3/B7 0.47-2.13μm ($w_1$) and of B3/B4 0.47-0.55μm ($w_2$=1- $w_1$) are functions of surface brightness expressed via uncertainty $\delta\tau^a$ (Eq. 6) as follows: $w_1$=1 if $0\le \delta\tau^a <0.05$ (dark surface); $w_1$=0 if $\delta\tau^a <0$ or $\delta\tau^a >0.5$ (bright surface); and linear function in between, $w_1$=( $\delta\tau^a$ -0.05)/0.45. The reflectance $\rho_\lambda(\tau^a)$ in Eq. (17) is LER (result of atmospheric correction) with AOD $\tau^a$. The minimization algorithm (17) incrementally increases AOD from the LUT until $F(\tau^a)$ reaches minimum, computes coefficients of quadratic polynomial based on 3 points encompassing the minimum, and analytically computes AOD in the minimum of quadratic function.

Our study of independent AOD retrievals using 0.47-0.55μm ratio (second term of Eq. 17) shows that it 1) has a reasonable accuracy over dark surfaces albeit somewhat lower than the standard algorithm (Eqs. 15-16); 2) is more stable over bright surfaces with zero or much lower positive AOD bias when atmospheric AOD is low; 3) underestimates AOD at high aerosol loading over all surfaces by as much as 20-50%. Given these properties, it is clear that the second term of Eq. 17, having lower sensitivity to AOD, mostly serves to stabilize solution over brighter surfaces under clean (low AOD) atmospheric conditions by minimizing high AOD bias from the first term.

When Smoke is detected, meaning that AOD is usually sufficiently high and effect of surface errors is reduced, we give more weight (if $w_1$<0.8 then $w_1$=0.8) to the standard retrieval (bands B3-B7) with much higher sensitivity to AOD.

Finally, when Dust is detected, the aerosol retrieval adds an additional term for the MODIS Red band B1, and uses equal weights for all three terms:

$$F(\tau^a) = (1 - R_{0.47}^T(\tau^a)/R_{0.47}^M)^2 + (1 - [\rho_{0.47}(\tau^a)/\rho_{0.55}(\tau^a)]/b_{34})^2 + (1 - R_{0.64}^T(\tau^a)/R_{0.64}^M)^2. \qquad (18)$$

The theoretical B1 TOA reflectance in the last term is computed using the accurate GF-solution (Eq. 2) with B1 BRDF model. As one can see from Table 1, properties of the dynamic dust model are such that the concentration of the coarse mode rapidly grows with AOD, increasing anisotropy of phase function and reducing backscattering. This reduction counteracts and slows down the respective increase of TOA reflectance. In effect, dynamic Model 6 requires a significantly higher AOD to match the measured reflectance at 0.47μm. We found experimentally that algorithm (17) significantly overestimates dust AOD by up to a factor of 2, however, adding the red-band term (Eq. 18) reduces AOD and significantly improves its accuracy. The mentioned spectral dis-balance of the dust Model 6 may be caused by our use of spheroidal model (Dubovik et al., 2006) to approximate dust particles. A similar spectral-angular mismatch from the use of spheroids to describe optics of the dust scattering was observed in analysis of MISR data (*personal communications with R. Kahn*).





Lastly, at high altitudes (H>4.2km, e.g. Tibetan plateau), AOD is not retrieved unless Smoke/Dust was detected. Our study shows that in conditions of very low AOD, non-flat terrain and generally bright surface, MAIAC aerosol retrievals at high altitudes are unreliable. Instead of retrievals, we assume a fixed climatology AOD$_{min}$=0.02 used for the atmospheric correction.

## 6.3 Bright Surface Bias Correction

Regardless of specific AOD retrieval algorithm, solution over bright surfaces can be unstable and can easily develop a positive bias. It should be mentioned that the term "bright surface" in MAIAC is understood in terms of low sensitivity ($\partial R / \partial \tau^a$ ~0) or high uncertainty ($\delta \tau^a$) of aerosol retrievals. The same surface can be "bright" in the backscattering directions, in particular close to the hot spot because of increase in the surface reflectance, and "dark" for the forward scattering geometries where the surface is considerably darker due to shadowing, in combination with higher aerosol phase function and the single scattering radiance. MAIAC retrievals show that AOD is systematically overestimated over some bright surfaces in the backscattering view directions correlating with the surface features, which is apparent in the time series of gridded AOD. These artefacts are generic and one can easily find them in the MODIS DT, DB, and in the VIIRS aerosol products. As MAIAC deals with the time series analysis of gridded data directly, we developed a special statistical correction procedure. It is designed to detect and minimize such spatially persistent bias, and is only applied in clear low AOD conditions to prevent cancelling the real aerosol signal. The idea is to look at the large area, evaluate an average AOD using the darkest pixels where solution can be trusted, and correct biased AOD over bright pixels with known history of bias using the area-average value.

The bright surface correction procedure is applied to mesoscale areas (150×150km$^2$), denoted as $\dot{R}$ below, and works as follows:

1) Compute average AOD for pixels in four bins of uncertainty: $\delta \tau^a$ ≤0.05f, 0.05< $\delta \tau^a$ ≤0.12f, 0.12< $\delta \tau^a$ ≤0.22f, 0.22< $\delta \tau^a$ ≤0.4f. The AOD retrieval is trustworthy in the first bin and usually in the second bin. The first two bins cover densely vegetated surfaces and dark soils, but extend to considerably (visually) brighter surfaces at low sun/view zenith angles and/or high atmospheric turbidity.

2) The AOD bias generally manifests itself as an increase in the average AOD with the bin number (uncertainty). In such case, we define the area-average value $\tau_{av}$ based on the first bin or the first two bins depending on statistics (the number of pixels in these bins), and set the high AOD threshold as *thresh*= $\tau_{av}$ +0.1.

3) Mask the pixel in the high bins 2-4 if its AOD ($\tau_{ij}$) exceeds the threshold. When pixel is masked, its cumulative bias counter (*q.indexHighBias*) is increased by 1, and the bias index for the current observation (*q.indexCurrentBias*) is set to 1.





4)   For pixel (i,j) with current and persistent high bias (*q.indexCurrentBias*=1 and *q.indexHighBias*>2), replace AOD with the value $\tau_{ij}w + \tau_{av}(1-w)$, where the weight increases along with the deviation of pixel's AOD from the average, $w=(\tau_{ij}/\tau_{av})^{-2} \leq 1$.

The above procedure is not applied when absorbing aerosols (Smoke/Dust) are detected, or when $\tau_{av}$>0.3 indicating

possibility of generally higher aerosol levels.

5)   Finally, it should be mentioned that the bias detection can be triggered randomly for almost any pixel, leading to accumulation of noise in the cumulative counter. For this reason, and to avoid cancelling the real aerosol signal over regular pixels, we compute the average bias detection noise over area Ṙ and subtract it from *q.indexHighBias* monthly, effectively zeroing it for the regular pixels.

It should be mentioned that the described algorithm is in large an empirical summary of numerous "trials and errors" using AERONET validation and minimization of systematic spatial/temporal artefacts as our main criteria.

## 7. Spatial AOD filtering and Smoothing

Cloud tests (sec. 4) were designed to capture reliable bright, cold, high, or spatially/spectrally contrasting clouds. Because natural transition from clear to cloudy is a continuum, we use two additional AOD-based filters to detect thin or sub-pixel

clouds at 1km resolution, all based on an assumption that aerosols have some degree of spatial homogeneity. The filters below are not used if absorbing aerosols (Smoke/Dust) were detected.

1) The first filter uses histogram-based technique following the DT algorithm (Levy et al., 2007) which applies it to the TOA reflectance in 20×20 500m pixels' boxes, filtering lower 20% and upper 50% of data as potentially affected by either shadows or clouds. The average reflectance of the remaining pixels is used for the DT AOD retrieval. In MAIAC, we apply a

similar technique to 25×25km$^2$ blocks using retrieved AOD and filtering high values only. The upper threshold is a function of the cloud fraction (CF) in the block, H=0.65–0.6CF/0.9, decreasing from 65% in cloud-free conditions to 5% when CF=0.9. The AOD threshold is defined as *thresh*=AOD$_H$+$\delta$, where $\delta$=0.2 when covariance is high (q.iFlag_HighCov=1), and $\delta$=0.1 otherwise. For pixels with AOD>*thresh*, the cloud mask value is set to possibly cloud, CM_PCLOUD.

This filter is not applied when absorbing aerosols (Smoke/Dust) were detected, as well as in clear (AOD$_{max}$<0.35) or

homogeneous (AOD$_{max}$- AOD$_{min}$ <0.2) conditions. Overall, this filter rather significantly improves the quality of the final AOD product.

In earlier versions, AOD for the filtered pixels was set to the FILL_VALUE. The current C6 version reports retrieved AOD for these pixels for possible applications as a research quality. The main impetus came from the air quality research groups: in particular, Drs. I. Kloog and A. Just showed that the histogram filter often cancels AOD retrievals over urban regions with

high aerosol spatial variability from human activity, e.g. south-western part of Mexico City. AERONET validation for Mexico City shows an improvement from added high AOD pixels which were previously mostly filtered as CM_PCLOUD.



A similar improvement was observed at Ispra (Italy) AERONET site, located on western edge of the Po Valley, Italy, where topography and proximity of aerosol emission sources create conditions for high spatial aerosol variability.

Most of CM_PCLOUD pixels are located in the transition zone from clear to confidently cloudy. The high 1km resolution AOD in this "twilight" region (Koren et al., 2007) may be useful to study the aerosol-cloud interactions. However, we should emphasize that for most applications, the user should only use the highest quality AOD with QA cloud mask value CM_CLEAR.

2) The second "spatial homogeneity" filter based on analysis of 3×3 pixels was proposed by *Emily et al*. (2011). We use it in the following form:

- find pixel with maximum AOD over 3×3 pixels (AOD$_{max}$);

- compute average AOD$_{av}$ in 3×3 area without this pixel;

- filter the maximum value (*CM_PCLOUD*) if AOD$_{max}$ > AOD$_{av}$ +0.2.

After detection of residual clouds with filters 1-2, a 3×3 running averaging window is applied to 1km AOD, except when smoke/dust were detected. The averaging serves to ameliorate residual errors of gridding which create noise in the surface SRC, BRDF and AOD. As this noise is local and spatially coherent (due to the systematic nature of MODIS orbits and footprint size/location with the scan angle), it is effectively reduced by the 3×3 smoothing filter.

## 8. Cloud Shadow Detection

With cloud detection completed, the next step 7) detects cloud shadows using geometric analysis (Simpson et al., 2000) and BRDF reduction test. Based on the view geometry, we generate the "line of shadow" for each cloudy pixel, which is a function of the cloud top height ($H_c$). Parameter $H_c$ is evaluated based on $Tb_{11}$-contrast between the cloud-free background ($q.Tb_s$) and a cloudy pixel assuming an average adiabatic lapse rate of 6.5° K/km ($H_c=(Tb_{11} - q.Tb_s)/6.5$). If the background brightness temperature cannot be reliably evaluated, we assume the maximal cloud height of 12km (Stubenrauch et al., 2010). The shadow is detected along the "line of shadow" if surface reflectance in the bright channels B2 (0.87μm) or B5 (1.24μm) falls below the BRDF-predicted value by a certain threshold. Based on visual evaluation, the developed approach captures up to 80-90% of cloud shadows.

## 9. Atmospheric Correction

Following cloud detection and aerosol retrieval, the atmospheric correction derives surface spectral BRF and BRDF model, and computes instantaneous albedo. The spectral BRF is derived for cloud-free and clear to moderately hazy (AOD$_{0.47}$<1.5) pixels by scaling pixel's BRDF in order to match the TOA reflectance (Lyapustin et al., 2012a). It uses Eq. 2 modified as follows:



$$R(\mu_0,\mu,\varphi) = R^A(\mu_0,\mu,\varphi) \;+\; cR^{Surf}(\mu_0,\mu,\varphi)\,, \tag{19}$$

where $R^{Surf}$ is a surface-reflected term computed using the current RTLS parameters and retrieved aerosol data, and $c$ is spectrally-dependent scaling factor. Then, the BRF is given by:

$$r_\lambda(\mu_0,\mu,\varphi) = c_\lambda \rho_\lambda(\mu_0,\mu,\varphi)\,, \tag{20}$$

where $\rho_\lambda$ is computed from the RTLS model for a given geometry. Because $R^{Surf}$ is a non-linear function of the surface reflectance, solving (19-20) takes 2 iterations. On the second iteration, the surface term $R^{Surf}$ is computed with scaled BRDF from the first iteration, $c_\lambda^{(1)}\rho_\lambda(\mu_0,\mu,\varphi)$, and the final scaling coefficient in (20) is a product $c_\lambda = c_\lambda^{(1)}c_\lambda^{(2)}$. Because the non-linearity is small, the second iteration has a very small effect on the final result (20).

In addition to 1 km, spectral BRF in MODIS bands 1-7 is also computed at 500m resolution nested in 1km grid. While MAIAC performs atmospheric correction of MODIS 250m bands (B1-B2) as well, these results are not currently included in the output files.

The RTLS model parameters are retrieved for bands B1-B8 only. Therefore, BRF in MODIS ocean bands B9-B12 is computed with the same approach (19-20) using BRDF from the nearest bands, for instance, B4 BRDF is used for B11-B12. The red band B1 BRDF (instead of B3) is used for AC of B9-B10 in the "blue" part of the spectrum. Over dense vegetation, the Red band reflectance is nearly as low as that in the Blue but it is significantly less affected by the aerosol retrieval errors. So, while the shapes of BRDF are very similar, it is somewhat more stable in the Red. At the same time, the difference in the magnitude of reflectance does not matter for the scaling approach (19-20) as long as the general BRDF shape is right in order to correctly model the surface reflection and upward propagation for the direct solar beam and diffuse (sky) irradiance.

A Lambertian assumption is used during the algorithm initialization period which may last from just 4 days (observations) in cloud-free low AOD conditions to over a month depending on cloudiness/snow cover. During this period of time, MAIAC performance is sub-optimal with higher rate of undetected clouds and reflectance biases from Lambertian assumption. In the ongoing C6 MODAPS processing of MODIS Terra and Aqua, MAIAC was initialized globally using the second half of 2002, and then the processing started from the beginning of 2000. In parallel, a separate forward processing stream using MODIS data from the second half of 2016 to initialize is expected to start soon.

The latest BRF combined with previous BRFs stored in the Queue are used for the BRDF inversion providing 3 parameters of the RTLS model. This represents a change from the original algorithm (Lyapustin et al., 2012a) which derived RTLS coefficients by matching the measured TOA reflectance. Using the original TOA measurements potentially allows to achieve a better accuracy of BRDF retrievals, however at expense of storing or re-computing a number of RT-functions for each past observation from the Queue. Our analysis showed that with the current good accuracy of MAIAC cloud detection and aerosol retrieval, the result of much simpler and faster BRF-based inversion is practically indistinguishable from the TOA-based inversion.



With C6+ calibration (Lyapustin et al., 2014) which added MODIS Terra polarization correction (Kwiatkowska et al., 2008), response-vs-scan (RVS) trending using quasi-stable desert calibration sites (Sun et al., 2014), and removed residual trends and cross-calibrated Terra to Aqua, we are using MODIS Terra and Aqua as one dataset. This doubles MODIS revisit frequency, a critical requirement for the time series analysis, which significantly helps MAIAC in all stages of processing, in particular for the BRDF retrieval and SRC characterization.

MAIAC has a surface change detection algorithm (Lyapustin et al., 2012a) based on analysis of geometrically normalized BRF in bands B1,B2,B7. For instance, normalization to SZA=45° and VZA=0° ($F_{0V}$=-0.0458621, $F_{0G}$=-1.1068192 in Eq. 1) at any wavelength uses the following formula (see Eq. (6) from Lyapustin et al., 2012a):

$$BRF_n = BRF * (k^L - 0.0458621*k^V - 1.1068192*k^G)/( k^L + F_V*k^V + F_G*k^G), \tag{21}$$

where $F_V$ and $F_G$ are volumetric and geometric kernels for the MODIS view geometry provided for the user's convenience in the MAIAC output (see sec. 101.2.2). Geometric (or BRDF-) normalization significantly (by a factor of 3-6) reduces BRF variations caused by the changing view geometry of MODIS with the orbit. MAIAC change detection looks for anti-correlated changes in the Red and NIR bands during the accumulation period, $\eta=\Delta BRF_n/BRF_{n,av}$, where $BRF_{n,av}$ is an average value. Based on this analysis, the surface state is characterized as stable or no change ($\eta<0.05$), and two categories of green-up or senescence, namely Regular change ($0.05<\eta<0.15$) and Big change ($\eta>0.15$, sec. 10.2.2).

Based on extensive empirical analysis, MAIAC undertakes RTLS inversion when the surface is relatively stable, $\eta<0.15$. When change is significant ($\eta\geq0.15$), the BRDF is scaled with the latest observation to adjust the total reflectance assuming the shape of BRDF does not change, as in (Schaaf et al., 2002). After inversion, the new BRDF goes through several tests verifying "correctness" of its shape, and its consistency with the previous solution stored in the Queue (for details, see (Lyapustin et al., 2012a)). In order to preserve consistency and reduce random noise, we are using update with relaxation,

$$\vec{K} = w\vec{K}^{new} + (1-w)\vec{K}^{prev}. \tag{22}$$

Above, the superscript indicates the new and previous solutions, and the weight $w$=0.5 when surface does not change, and $w$=0.7 for the regular change. While such update practice generally improves quality of the BRDF model during stable periods, it delays the BRDF model response to the surface change in addition to the delay related to the length of the Queue. On the contrary, spectral $BRF_n$ represents an instantaneous surface snapshot from the latest observation. For this reason, studies of vegetation phenology, seasonality etc. should use $BRF_n$ rather than BRDF model-based reflectance values.

Many applications, including higher-level algorithms for vegetation characterization, e.g., LAI/FPAR (Chen et al., 2017), global model assimilation etc., require knowledge of uncertainty. We provide *BRF uncertainty (Sigma_BRFn)* in MODIS Red (B1) and NIR (B2) bands at 1km defined as a standard deviation of the BRFn over the accumulation period of the Queue (4-16 days) under assumption that the surface is stable or changes linearly in time. This is one of the most conservative and realistic estimates of uncertainty which includes contribution from gridding, undetected clouds, errors of atmospheric correction including those from the aerosol retrieval, and of surface change when reflectance change is non-





linear over the length of the Queue. *Sigma_BRFn* in the Red band can serve as a proxy of uncertainty at shorter wavelengths, where the surface is generally darker, and the NIR value can be a proxy for the longer wavelengths with high surface reflectance.

With detection of snow, MAIAC "freezes" the land spectral BRDF in the Q-memory, and switches to the snow processing
mode retrieving sub-pixel snow fraction and snow grain size. The total surface reflectance (albedo) in this case is computed as a linear mixture of land BRDF and snow reflectance given by a semi-analytical model (Lyapustin et al., 2010).

## 10. MCD19 Data Products and Quality Assurance

MAIAC provides a suite of MODIS atmospheric and surface products in three HDF4 files: *daily* MCD19A1 (spectral BRF, or surface reflectance), *daily* MCD19A2 (atmospheric properties), and *8-day* MCD19A3 (spectral BRDF/albedo). As this
paper describes the first official public release of MAIAC MODIS data, we consider it useful to provide a brief technical description of MAIAC products and its quality assurance flags (QA) which is given in the User's Guide in more detail.

### 10.1 Tiled File Structure and Naming Convention

All products are reported on 1km sinusoidal grid. The sinusoidal projection is not optimal due to distortions at high latitudes and off the grid-center, but it is a tradeoff made by the MODIS land team for the global data processing. The gridded data
are divided into 1200×1200km$^2$ standard MODIS tiles shown in Figure 5. The current dataset presents data per orbit. Each daily file name follows the standard MODIS name convention, for instance:

MCD19A1.DayOfObservation.TileNumber.Collection.TimeOfCreation.hdf.    DayOfObservation    has    the    format "YYYYDDD", where YYYY is year, DDD is Julian day. TileNumber has the standard format, e.g. h11v05 for the east coast USA.
Each daily file usually contains multiple orbit overpasses (1-2 at equator and up to 30 in the polar regions for combined Terra and Aqua) which represents the third (time) dimension of MAIAC daily files. The orbit number and the overpass time of each orbit are saved in global attributes "*Orbit_amount*" and "*Orbit_time_stamp*" sequentially. The Orbit_time_stamp has the format "YYYYDDDHHMM[TA]", where YYYY is year, DDD is Julian day, HH is hour MM is minute, and T or A stand for Terra or Aqua. At high latitudes, only the first 16 orbits with largest coverage are selected for processing per day in
order to limit the file size.

### 10.2 MAIAC Products: General Description

MAIAC conducts processing over global land tiles and land-containing ocean tiles (green and light blue colors in Fig. 5). Over inland, coastal and open ocean waters, MAIAC reports AOD, fine mode fraction, and spectral reflectance of underlight (water-leaving radiance). MAIAC processing over water will be described in a separate publication.



### 10.2.1 Atmospheric Properties File (MCD19A2)

For each orbit, MAIAC *daily* MCD19A2 (atmospheric properties) file includes the following parameters listed in Table 2a. Over land, MAIAC reports the following parameters at 1km resolution: AOD at 0.47 and 0.55 μm and AOD uncertainty evaluated using Eq. (9) for cloud-free and ***possibly cloudy*** pixels; column water vapor (cm) for all pixels; Injection Height of
Smoke plume (in m above ground); background aerosol model used in the retrievals (see Fig. 4). The aerosol type (result of Smoke/Dust test) is reported in QA bits 13-14 (Aerosol Model) of Table 2b.

Over water, we report AOD outside of the glint area. Current processing has a glint angle cutoff of $\leq 40°$ as in the DT over ocean algorithm (Levy et al., 2013). When MAIAC detects dust, AOD is also reported for smaller glint angles when measured TOA reflectance at 1.24μm (B5) significantly exceeds reflectance from the ocean surface predicted by the Cox-
Munk model (Cox and Munk, 1954) for a given wind speed. Over the open ocean and large in-land lakes (e.g. Great Lakes of North America), we also report Fine Mode Fraction (FMF). FMF is not retrieved over small in-land water bodies.

In addition to the Blue band AOD (0.47μm), MAIAC also reports AOD at 0.55μm which is computed based on spectral properties of the aerosol model used in retrievals. It is provided to support the regional and global chemical transport and climate simulation models, AOD validation and AOD product intercomparison, all standardized to 0.55μm. Validation
shows that quality of AOD at 0.55μm is generally close though slightly worse than the original retrieval at 0.47μm.

The QA structure for MCD19A2 file is presented by Table 2b.

### 10.2.2 Surface Reflectance File (MCD19A1)

For each orbit, MAIAC daily MCD19A1 (surface reflectance) file includes parameters shown in Table 3a.

Over cloud-free land and clear-to-moderately turbid ($AOD_{0.47}<1.5$) conditions, for solar zenith angles below 80°, file
MCD19A1 reports surface BRF at 1km in bands 1-12, and at 500m in bands 1-7; 1km BRF uncertainty (*Sigma_BRFn*) in MODIS Red (B1) and NIR (B2) bands. When snow is detected, we report snow grain size (diameter, in mm), sub-pixel snow fraction, and *rmse* (*Snow_Fit*) between MODIS measurements in bands B1, B5, B7 and the linear mixture model of spectral snow reflectance and land spectral BRDF at 1km. Following the Sun-View geometry suite at 5km, MCD19A1 also reports values of volumetric ($F_v$) and geometric-optics ($F_g$) kernels of RTLS model for the geometry of observation. The kernels are
provided for the ease of users' geometric- (or BRDF-) normalization of spectral BRFs using Eq. 21. One can easily modify normalization to a preferable Sun angle according to latitude or season, by replacing coefficients in the numerator of Eq. 21 with values from Table 1 in the User's Guide calculated for different solar zenith angles and nadir view.

Over water, MCD19A1 reports diffuse reflectance of underlight (of water-leaving radiance) in bands 1-12.

Table 3b shows QA definition for the surface reflectance file. Bits 0-2, 3-4 and 5-7 are the same as in Table 2b. The QA bits
8 and 9 carry an additional information about quality of atmospheric correction. For instance, better quality is achieved at low AOD and when the surface BRDF is known (Algorithm is initialized) as opposed to high AOD and/or "Not initialized" status when Lambertian assumption is used in the atmospheric correction.



### 10.2.3 Surface BRDF File (MCD19A3)

The 8-day MCD19A3 (BRDF/albedo) file reports parameters of RTLS BRDF model ($k_L$, $k_v$, $k_G$) for MODIS bands B1-B8, number of days since the last RTLS model update (Update_Day), and instantaneous surface albedo for the overpass time in bands 1-8 at 1km resolution. These parameters are listed in Table 4.

### 10.3 Quality Assurance

In *daily* output files, the QA reports cloud mask, adjacency mask, surface type (the result of MAIAC dynamic Land-Water-Snow classification), and a surface change mask. In general, MAIAC aerosol-surface retrievals are only performed for cloud-free pixels (QA.Cloud_Mask = Clear) except AOD which is also reported for the value *Possibly Cloudy*. As discussed in sec. 7, this AOD may be used with caution in specific well understood cases, e.g. at high spatial variability of aerosol or aerosol analysis near clouds. Because most of pixels with QA.Cloud_Mask = *Possibly Cloudy* contain residual cloud contamination, these pixels are not recommended for the general use.

Adjacency mask gives information about detected clouds or snow in the ±2-pixel vicinity. For most applications, we recommend to only use data with QA.AdjacencyMask=Clear (000). The value 011 (Adjacent to a single cloudy pixel) can also be used as it often represents a false cloud detection. The other categories of Q.AdjacencyMask are not recommended when using either AOD or BRF products because neighbor clouds or snow increase possibility of residual cloud/snow contamination of a given pixel, resulting in overestimation of AOD and respective errors of atmospheric correction.

To select the best quality AOD, one should use QA.QA_AOD = Best_Quality which combines the best values of cloud and adjacency masks: QA.CloudMask=Clear and QA.AdjacencyMask=Clear.

For the best quality BRF, one should apply the following QA filter:

QA.AODLevel=low (0), QA.AdjacencyMask=Clear, and QA.AlgorithmInitializeStatus= initialized (0).

We should admit that the current QA structure is not optimal and may be improved in the future.

### 10.4 Known Issues and Limitations

Below, is a list of currently known issues and limitations of algorithm MAIAC:

1. The maximum value of LUT $AOD_{0.47}$ is 4.0 which limits characterization of strong aerosol emissions.
2. MAIAC LUTs are built assuming pseudo-spherical correction in single scattering which has a reduced accuracy for high sun/view zenith angles. A reduced MAIAC performance is expected at solar zenith angles > 70°.
3. MAIAC may be missing bright salt pans in several world deserts. In such cases, it generates a persistent high AOD resulting in missing surface retrievals.
4. Geographic AOD boundaries may sometimes be observed on borders of the regional aerosol models when they have a significant difference in absorption.



5. Because of inherent uncertainties of gridding on the coastline, the area of ±1-3 pixels from the coastline may contain frequent artifacts in cloud mask (usually over-detection), AOD (higher values) and surface BRF. Users should exercise caution near the coastline as indicated by QA.QA_AOD (value 1010).

6. AC over detected snow: as MAIAC does not retrieve AOD over snow, it assumes a low climatology AOD=0.05 globally and 0.02 at high elevations (H>4.2km). Over north-central China, which is often heavily polluted and low AOD assumption can lead to a significant bias, we use AOD averaged over mesoscale area of 150km using reliable AOD retrievals over snow-free pixels. Such approach does improve quality of AC as compared to low-AOD assumption as judged by the reduced boundaries and diminished color artefacts, but it does not account for the aerosol variability inside 150km area which may be significant.

7. Ice mask is currently unreliable.

8. Consistently miss a particular type of clouds (moderately thin and homogeneous cumulus) over water generating high AOD.

9. MAIAC uses a specialized "Bay" mask for aerosol retrievals over coastal waters with high sediments. The current "Bay" mask misses several such areas where AOD retrievals often show a high bias.

## 10.5 An Example of MAIAC Products

To illustrate MAIAC product suite, Figure 6 shows the global daily-composite browse images at 20km resolution for selected products including AOD (0-2), column water vapor (0-5cm), RGB BRF, snow fraction (0-1) and RGB of the isotropic parameter ($k^L$) of the RTLS model which gives indication of spectral BRDF and serves as a proxy of the general surface brightness and spectral albedo. The numbers in parenthesis give the scale range. The browse images were generated by the MODAPS Land processing team (Roy et al., 2002) as part of the product quality evaluation.

The browse images are shown for days 60 and 230 of 2005: day 60 shows a considerable snow cover in the northern hemisphere in RGB BRF with the corresponding high snow fraction; dust storms in the northern Sahara and in Taklimakan desert, and high AOD levels in the Indo-Gangetic plane, northern China and south-East Asia. In contrast, on day 230 cloud-free observations show detected snow only over Greenland and polar North, as well as southern Andes. AOD shows strong forest fires in Alaska and large-scale biomass burning in southern Amazon with smoke transported south-east across South America. It also reveals dust storms in Western Sahara and Thar desert, and high aerosol levels in Southern Africa. The RGB of RTLS $k^L$ parameter is naturally gap-filled, and shows contrasting seasonal dynamics of vegetation between the northern and southern hemispheres. The column water vapor shows seasonal, latitudinal and vertical variations, the latter associated with retrievals above the clouds.

Finally, Figure 7 presents results of the global MAIAC AOD validation against AERONET (Holben et al., 1998) showing correlation coefficient, average bias, and *rmse* for individual AERONET sites along with the global scatterplot during 2000-2016. The detailed validation analysis of MAIAC dataset, and its comparison with the standard products from MODIS or other sensors deserves a separate consideration, so this analysis merely serves to illustrate the overall quality of MAIAC



aerosol retrievals. Figure 7 shows *a)* predominantly high correlation with AERONET except for the world regions where typically both AOD and its range of variation are low (e.g., south-western USA or south of South American continent); *b)* globally low bias and *rmse* except major biomass burning, industrial or mineral dust source regions such as Sahara, Sahel and sub-tropical Africa, Indo-Gangetic Plane, south Asia and China. The higher *rmse* in these source regions is typical of all aerosol retrieval products and is expected due to high variability of aerosol types and properties, often in combination with the bright land surface increasing uncertainties of satellite retrievals. The bias shows clustering of results, and gives a clear indication for the required tuning of MAIAC regional aerosol models, e.g. in South Asia and China. Some of these biases come from the seasonal variation in aerosol properties (e.g., Mhawish et al., in review) which will be implemented in the next version of MAIAC.

## 11. Conclusion

This paper presented the C6 MAIAC algorithm used in the ongoing MODIS Collection 6 processing. MAIAC cloud detection, aerosol retrieval and atmospheric correction over land were described in detail. Being the first publication of the official new MODIS product MCD19, this paper also provided technical specification of MCD19 output files along with the brief quality assurance discussion and recommendations for use. Other MAIAC components related to detection and processing of snow, retrievals over water and smoke plume height retrieval will be described elsewhere.

The paper also presented a brief analysis of near-global MAIAC AOD validation against AERONET measurements. These results serve to complement the growing body of the air quality and land community analyses on MAIAC data quality and its comparison to the standard MODIS products.

**Acknowledgements**. The research of A. Lyapustin, Y. Wang and S. Korkin was funded by the NASA Science of Terra, Aqua, SNPP (17-TASNPP17-0116; solicitation NNH17ZDA001N-TASNPP). A. Lyapustin was additionally supported by the NASA GeoCAPE program. The work of D. Huang was funded by the NASA DSCOVR program. We appreciate a large work of MODAPS team on MAIAC integration, in particular support of E. Masuoka and S. Devaldiga, and support of LP DAAC. The lasting support of the NASA Center for Climate Simulations in continental-scale testing and multiple internal releases of MAIAC data has been invaluable. We are grateful to the AERONET team for providing validation data. Lastly, we would like to express gratitude to multiple users and user groups in the land and air quality communities whose continuous analysis of MAIAC MODIS data helped bring MAIAC to its current level.

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



**Author Contribution:** A. Lyapustin developed code MAIAC and prepared the manuscript. Y. Wang developed operational version of MAIAC and conducted global evaluation and testing. S. Korkin supported computation of MAIAC look-up tables. D. Huang conducted global validation of MAIAC aerosol product.

5   **Competing interests:** The authors declare that they have no conflict of interest.





**List of Tables:**

**Table 1.** Microphysical properties of MAIAC aerosol models: radius and standard deviation of fine and coarse fractions of bi-lognormal volume size distribution; ratio of volume concentrations (coarse to fine) as functions of AOD ($\tau$); real and imaginary refractive index ($n=m\text{-}ik$); Angstrom (AAE) parameter for $k$ ($k(\lambda) = k(\lambda_0)(\lambda/\lambda_0)^{-b}$ for $\lambda<\lambda_0=0.66$ μm, and $k(\lambda) = k(\lambda_0)$ for $\lambda>\lambda_0$). Finally, the last column shows fraction of spherical particles where 1 represents spheres and 0 represent spheroids from the DLS model (Dubovik et al., 2006).

**Table 2a.** Reported parameters in Atmospheric Properties File (MCD19A2).

**Table 2b.** AOD QA definition for MCD19A2 (16-bit unsigned integer)

**Table 3a.** Reported parameters in Surface Reflectance File (MCD19A1)

**Table 3b.** Surface reflectance QA definition for MCD19A2 (16-bit unsigned integer).

**Table 4.** 8-day BRDF model parameters (MCD19A3)





| Model | $R_v^F$ , µm | $\sigma_v^F$ , µm | $R_v^C$ , µm | $\sigma_v^C$ , µm | $C_v^C / C_v^F$ | $M$ | $k_{0.66}$ | AAE | Mie-Frac |
|---|---|---|---|---|---|---|---|---|---|
| 1 | 0.12+0.05τ ≤0.2 | 0.35+0.05τ ≤0.45 | 2.8+0.2τ ≤3.2 | 0.6+0.1τ ≤0.8 | 0.6 | 1.42 | 0.0045 | 0 | 1 |
| 2 | 0.16 | 0.4 | 2.4 | 0.6 | 0.5 | 1.48 | 0.0035 | 0 | 0.8 |
| 3 | 0.13 | 0.5 | 2.8 | 0.7 | 1 | 1.48 | 0.012 | 0 | 0.6 |
| 4 | 0.12+0.05τ ≤0.2 | 0.35+0.05τ ≤0.45 | 2.8+0.2τ ≤3.2 | 0.6+0.1τ ≤0.8 | 0.6 | 1.42 | 0.0065 | 0 | 1 |
| 5 | 0.15+0.05τ ≤0.2 | 0.45+0.1τ ≤0.55 | 2.5+0.3τ ≤2.8 | 0.6+0.1τ ≤0.8 | 1.4 | 1.44 | 0.005, 0.67µ | 0.5 | 0.9 |
| 6 | 0.12 | 0.5 | 1.9 | 0.6 | 0.02(1+τ) /0.9τ | 1.56 | 0.001, 0.67µ | 2.0 (cur) | 0 |
| 7 | 0.12+0.025τ ≤0.2 | 0.4 | 3.2+0.2τ ≤3.8 | 0.7 | 0.7 | 1.51 | 0.009 | 0 | 1 |
| 8 | 0.15+0.05τ ≤0.2 | 0.45+0.1τ ≤0.55 | 2.5+0.3τ ≤2.8 | 0.6+0.1τ ≤0.8 | 1.4 | 1.44 | 0.0065, 0.67µ | 0.5 | 0.9 |

**Table 1**. Microphysical properties of MAIAC aerosol models: radius and standard deviation of fine and coarse fractions of bi-lognormal volume size distribution; ratio of volume concentrations (coarse to fine) as functions of AOD ($\tau$); real and imaginary refractive index ($n=m-ik$); Angstrom (AAE) parameter for $k$ ($k(\lambda) = k(\lambda_0)(\lambda/\lambda_0)^{-AAE}$ for $\lambda<\lambda_0=0.66$ µm, and $k(\lambda) = k(\lambda_0)$ for $\lambda>\lambda_0$). Finally, the last column shows fraction of spherical particles where 1 represents spheres and 0 represent spheroids from the DLS model (Dubovik et al., 2006).




| SDS name | Scale | Description |
|---|---|---|
| Optical_Depth_047 | 0.001 | Blue band aerosol optical depth |
| Optical_Depth_055 | 0.001 | Green band aerosol optical depth |
| AOD_Uncertainty | 0.0001 | AOD uncertainty |
| FineModeFraction | 0.0001 | Fine mode fraction over water |
| Column_WV | 0.001 | Column Water Vapor (cm) |
| Injection_Height | n/a | Smoke injection height (m above ground) |
| AOD_QA | n/a | AOD QA |
| AOD_MODEL | 0.001 | Regional background model used |
| Sun – View Geometry Suite at 5km | | |

The 5km Sun-View geometry suite includes:

| cosSZA | 0.0001 | Cosine of solar zenith angle (5km) |
|---|---|---|
| cosVZA | 0.0001 | Cosine of view zenith angle (5km) |
| RelAZ | 0.01 | Relative azimuth angle (5km) |
| Scattering_Angle | 0.01 | Scattering angle (5km) |
| Glint_Angle | 0.01 | Glint angle (5km) |

5  **Table 2a.** Reported parameters in Atmospheric Properties File (MCD19A2).





| Bits | Definition |
|---|---|
| 0-2 | **Cloud Mask**<br><br>000 --- Undefined<br><br>001--- Clear<br><br>010 --- Possibly Cloudy (detected by AOD filter)<br><br>011 --- Cloudy (detected by cloud mask algorithm)<br><br>101 --- Cloud Shadow<br><br>110 --- Hot spot of fire<br><br>111 --- Water Sediments |
| 3-4 | **Land Water Snow/Ice Mask**<br><br>00 --- Land<br><br>01 --- Water<br><br>10 --- Snow<br><br>11 --- Ice |
| 5-7 | **Adjacency Mask**<br><br>000 --- Clear<br><br>001 --- Adjacent to clouds<br><br>010 --- Surrounded by more than 8 cloudy pixels<br><br>011 --- Adjacent to a single cloudy pixel<br><br>100 --- Adjacent to snow<br><br>101 --- Snow was previously detected in this pixel |
| 8-11 | **QA for AOD over Land and Water**<br><br>0000 --- Best quality<br><br>0001 --- Water Sediments are detected (water)<br><br>0011 --- There is 1 neighbor cloud<br><br>0100 --- There is >1 neighbor clouds<br><br>0101 --- no retrieval (cloudy, or whatever)<br><br>0110 --- no retrievals near detected or previously detected snow<br><br>0111 --- Climatology AOD (0.02): altitude above 4.2km (Land)/3.5km (water)<br><br>1000 --- No retrieval due to sun glint over water<br><br>1001 --- Retrieved AOD is very low (<0.05) due to glint (water)<br><br>1010 --- AOD within +-2km from the coastline (may be unreliable) |





| | | |
|---|---|---|
| | 1011 --- Land, research quality: AOD retrieved but CM is possibly cloudy | |
| 12 | **Glint Mask** | |
| | 0 --- no glint | |
| | 1 --- glint (glint angle < 40°) | |
| 13-14 | **Aerosol Model** | |
| | 00 --- Background model (regional) | |
| | 01 --- Smoke detected | |
| | 10 --- Dust model (dust detected) | |
| 15 | **Reserved** | |

**Table 2b.** AOD QA definition for MCD19A2 (16-bit unsigned integer).





| SDS name | Scale | Description |
|---|---|---|
| Sur_refl[1-12] | 0.0001 | Surface reflectance, bands 1-12 |
| Sigma_BRFn[1-2] | 0.0001 | BRFn uncertainty, bands 1-2 |
| Snow_Fraction | 0.0001 | Snow fraction |
| Snow_Grain_Size | 0.001 | Snow grain diameter (mm) |
| Snow_Fit | 0.0001 | Land-Snow Mixture model RMSE in bands 1,5,7 |
| Status_QA | n/a | QA bits |
| Sur_refl_500m[1-7] | 0.0001 | Surface reflectance at 500m, bands 1-7 |
| Sun – View Geometry Suite (5km) | | |
| $F_v$ | n/a | RTLS volumetric kernel (5km) |
| $F_g$ | n/a | RTLS geometric kernel (5km) |

**Table 3a.** Reported parameters in Surface Reflectance File (MCD19A1).



| Bits | Definition |
|------|------------|
| 0-2 | **Cloud Mask** |
| 3-4 | **Land Water Snow/Ice Mask** |
| 5-7 | **Adjacency Mask** |
| 8 | **AOD level**<br><br>0 --- AOD is low (<=0.6)<br><br>1 --- AOD is high (> 0.6) or undefined |
| 9 | **Algorithm Initialization Status**<br><br>0 --- Algorithm is initialized<br><br>1 --- Algorithm is not initialized |
| 10 | **BRF retrieved over snow assuming AOD = 0.05**<br>0 --- no<br>1 --- yes |
| 11 | **Altitude >4.2km (land)/3.5km (water), BRF is retrieved using climatology AOD =0.02**<br>0 --- no<br>1 --- yes |
| 12-14 | **Surface Change Mask**<br><br>000 --- no change<br><br>001 --- Regular change: Green up<br><br>010 --- Big change: Green up<br><br>011 --- Regular change: Senescence<br><br>100 --- Big change: Senescence |

**Table 3b.** Surface reflectance QA definition for MCD19A2 (16-bit unsigned integer).



| SDS name | Scale | Description |
|---|---|---|
| Kiso | 0.0001 | RTLS isotropic weight, bands 1-8 |
| Kvol | 0.0001 | RTLS volumetric weight, bands 1-8 |
| Kgeo | 0.0001 | RTLS geometric weight, bands 1-8 |
| Sur_albedo | 0.0001 | Surface albedo, bands 1-8 |
| Update_Day | n/a | Number of days since the last update |

**Table 4.** 8-day BRDF model parameters (MCD19A3).





**List of Figures:**

**Figure 1**. Block-diagram of MAIAC algorithm.

**Figure 2**. Illustration of MAIAC time series processing for the mid-Atlantic USA 250km region with the New York City in

5    the low left corner. The rows show MODIS observations for different days of the year (DOY) for 2012. The two bottom

rows show DOY 314 from Terra and Aqua. The columns present MODIS TOA RGB image, MAIAC products (cloud mask,

$AOD_{0.47}$, RGB BRF, column water vapor) and some of the internal fields used in the processing ("deviation from clear-sky"

$\delta_{0.47}$, "cirrus band" reflectance $R_{1.38}$, thermal contrast, $dTb_{4\text{-}11}$ and its atmospheric part, $dTb_{4-11}^{A}$). Columns 3, 5-9 are

displayed using the rainbow palette with the (min-max) values shown in the heading in parentheses. The cloud mask uses the

10   following legend: cloud (red), possibly cloudy (yellow), cloud shadow (dark red), clear land (blue), clear snow (white), clear

water (light blue), clear water, detected sediments (grey), glint over water (dark grey).

**Figure 3**. Schematic illustration of MAIAC dynamic SRC retrievals featuring two independent lines of update, $b_1$ and $b_2$.

15   **Figure 4**. Map of background regional aerosol models specified in Table 1. The transparent yellow shape approximates the

dust regions.

**Figure 5.** Illustration of MODIS tiles for the sinusoidal grid. MAIAC performs processing over green and light blue (land-

containing) tiles.

**Figure 6.** Global browse images showing MAIAC AOD, column water vapor, RGB BRF, snow fraction and RGB of the

isotropic parameter ($k^L$) of the RTLS model for days 60 (top row) and 230 (bottom row) of 2005.

**Figure 7.** Results of global MAIAC AOD validation against AERONET from MODIS Terra and Aqua 2000-2016 record.



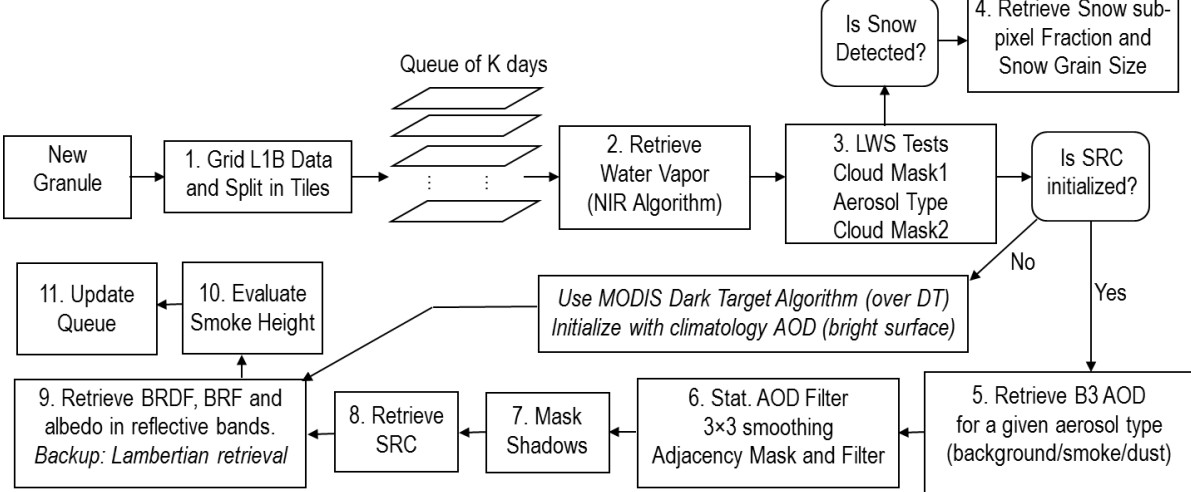

Figure 1. Block-diagram of MAIAC algorithm.





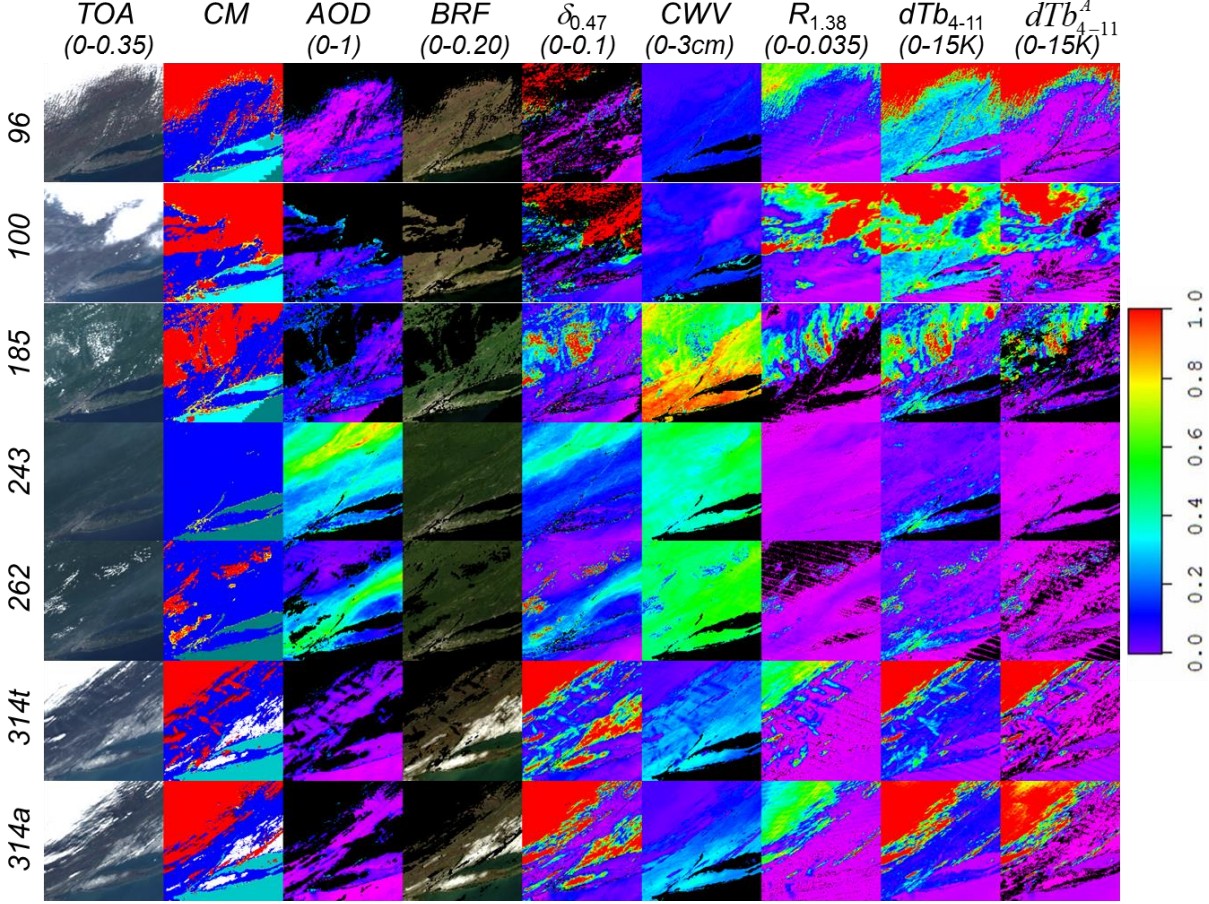

**Figure 2**. Illustration of MAIAC time series processing for the mid-Atlantic USA 250km region with the New York City in the low left corner. The rows show MODIS observations for 7 different days of the year (DOY) for 2012. The two bottom rows show DOY 314 from Terra and Aqua. The columns show MODIS measurements (TOA RGB image), MAIAC products (cloud mask, $AOD_{0.47}$, RGB BRF, column water vapor) and some of the internal fields used in the processing ("deviation from clear-sky" $\delta_{0.47}$, "cirrus band" reflectance $R_{1.38}$, thermal contrast, $dTb_{4-11}$ and its atmospheric part, $dTb_{4-11}^{A}$). Columns 3, 5-9 are displayed using the rainbow palette with the (min-max) values shown in the heading in parentheses. The cloud mask uses the following legend: red (cloud), yellow (possibly cloudy), dark red (cloud shadow), blue (clear land), white (clear snow), light blue (clear water), grey (clear water, detected sediments), dark grey (glint, water).






**Figure 3**. Schematic illustration of MAIAC dynamic SRC retrievals featuring two independent lines of update, $b_1$ and $b_2$.

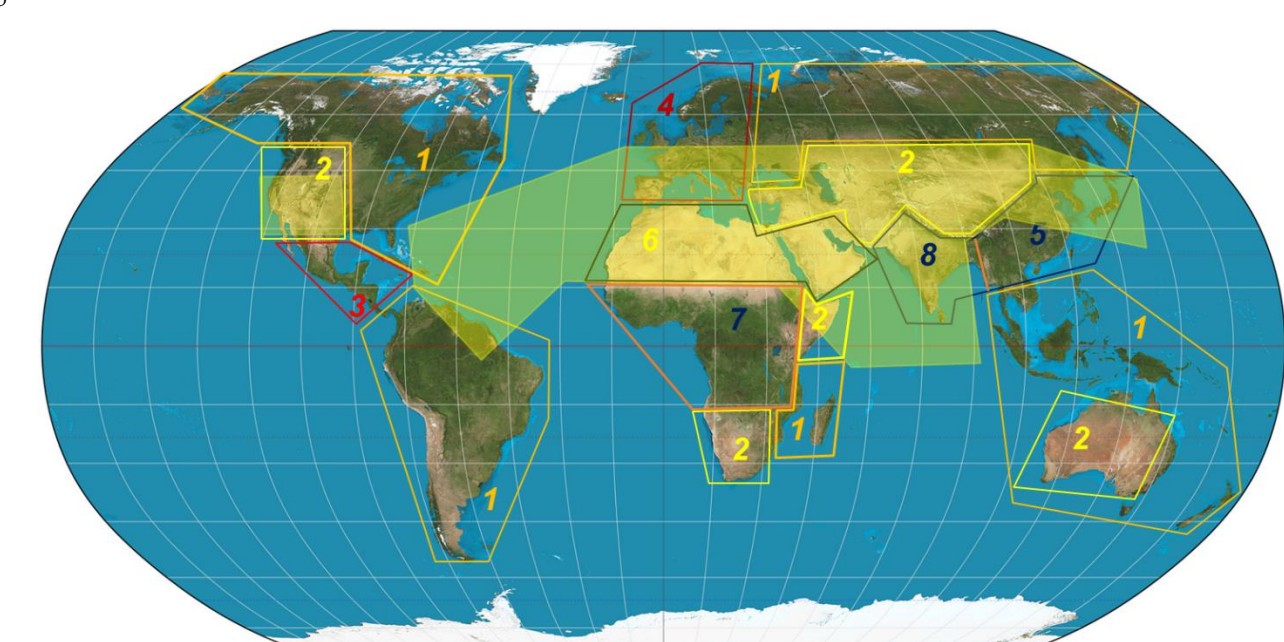

**Figure 4**. Map of background regional aerosol models specified in Table 1. The transparent yellow shape approximates the dust regions.





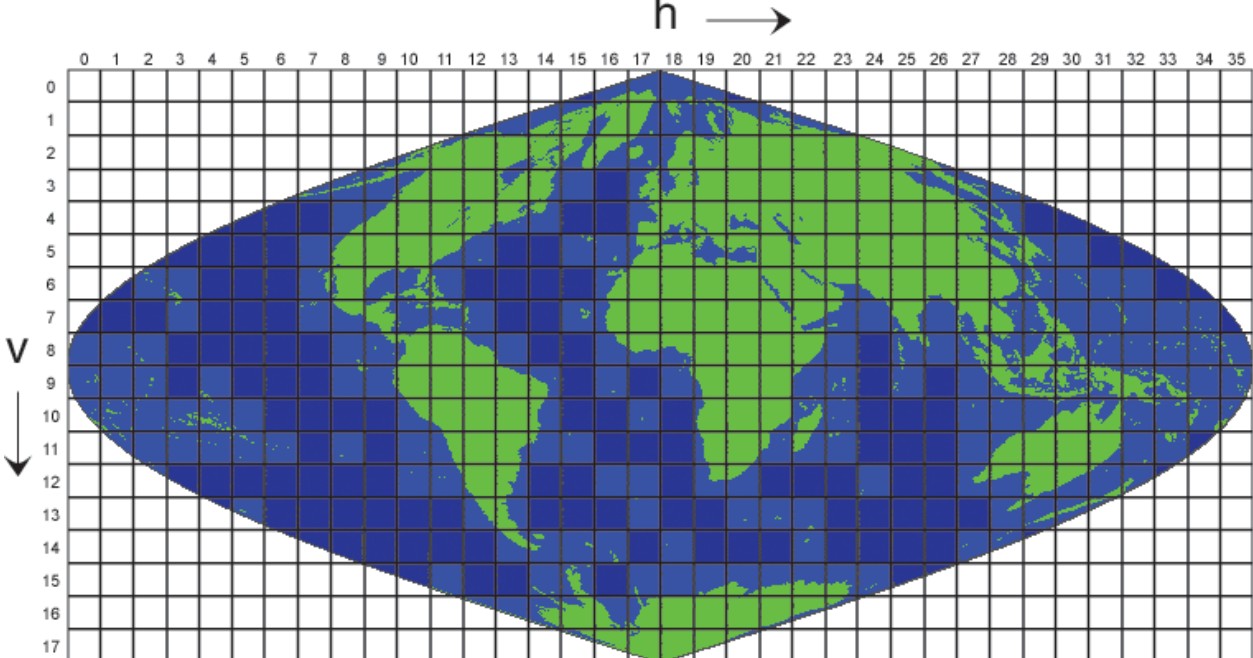

5    **Figure 5.** Illustration of MODIS tiles for the sinusoidal grid. MAIAC performs processing over green and light blue (land-containing) tiles.





**Figure 6.** Global browse images showing MAIAC AOD (scale 0-2), column water vapor (scale 0-5cm), RGB BRF, snow
fraction (scale 0-1) and RGB of the isotropic parameter ($k^L$) of the RTLS model for days 60 (top row) and 230 (bottom row)
10    of 2005.





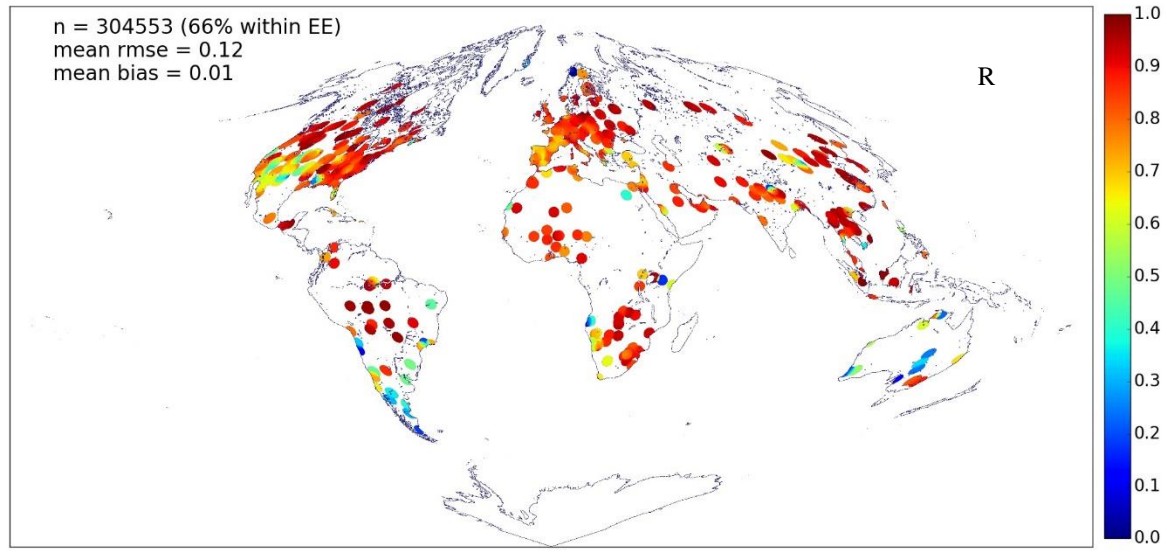

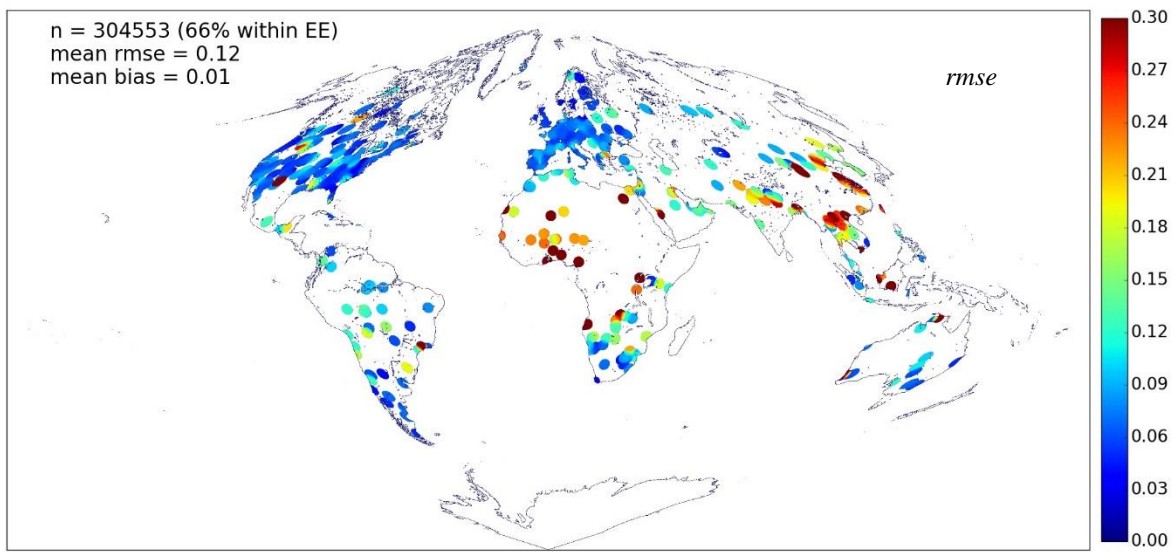





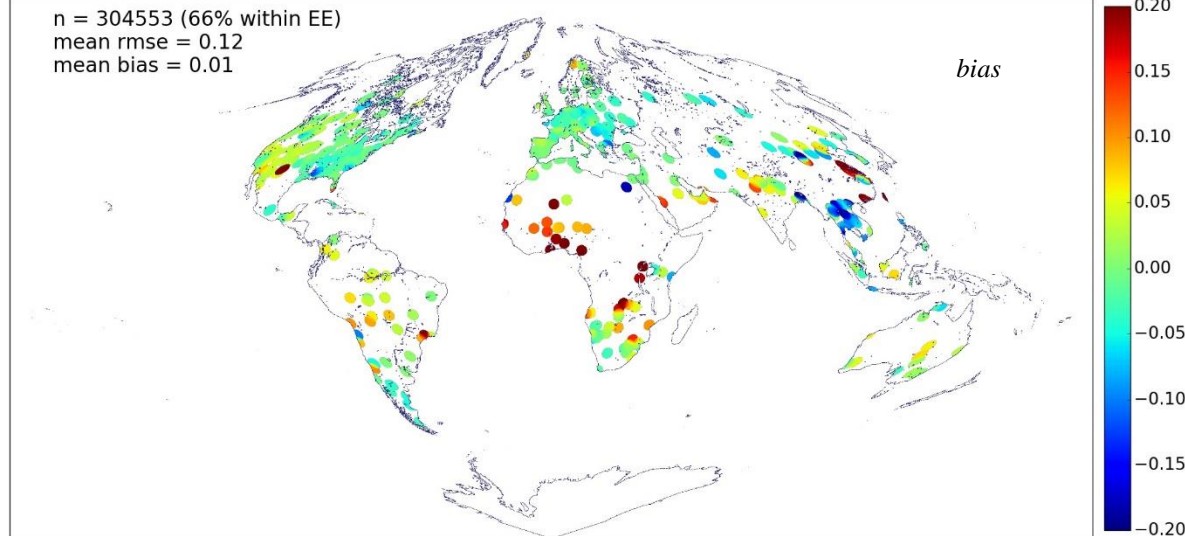





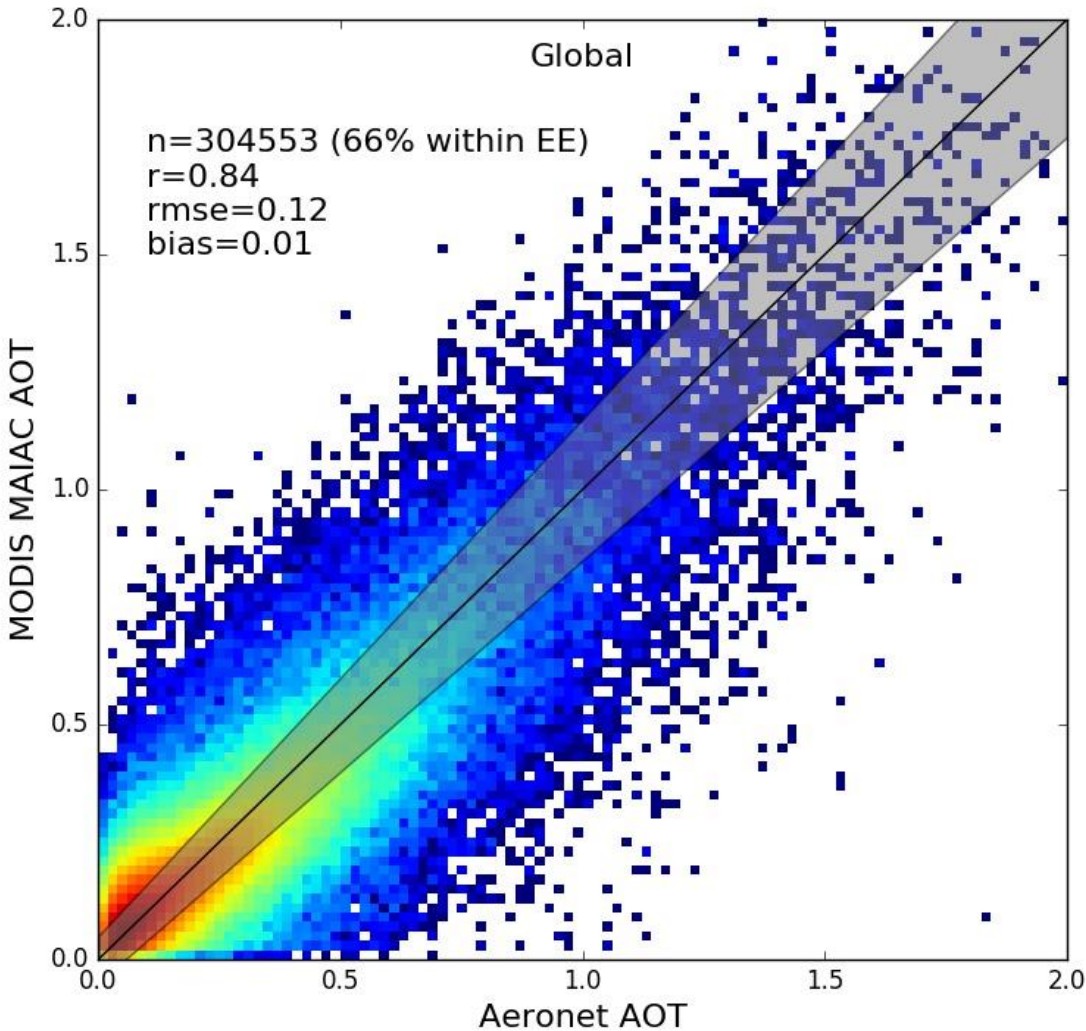

**Figure 7.** Results of global MAIAC AOD validation against AERONET from MODIS Terra and Aqua 2000-2016 record
(correlation coefficient, *rmse*, bias, and global scatterplot).