# Peer review of "MODIS Collection 6 MAIAC Algorithm"

_Atmospheric Measurement Techniques, 2018_

## Referee Comment (RC1) · Anonymous Referee #1 · 21 Jul 2018

Dear Authors,

First, thank you for a well-written detailed manuscript. The MAIAC product you describe is a very significant addition to the Earth system monitoring capability enabled by the MODIS instruments. I have only a few clarifying areas and this manuscript can be ready for publication. I have numerous smaller suggested edits below, and then several typographic edits at the bottom.

1) You should be congratulated for the "Known Issues and Limitations" section, this is very valuable information to compile and include. No one knows this product better than the authors, and this information is very valuable to end users even though it is not fully digested (if it were fully digested it would probably be solved!)

2) The assumed background AOD seems very important. If the assumed background is too low by 0.01, what is the resulting error in the retrieved AOD?

[Figure]

3) If the "reference SRC" is updated before retrieval of AOD, this means it is impossible for MAIAC to retrieve an AOD lower than the assumed background. Does the two-stream method described for correcting the "reference SRC" upward mitigate the upward trend in AOD that would be expected from downward updates to the "reference SRC"?

MINOR notes and edits: Page-Line

19-20 How are cloud shadows flagged in the output product?

4-15 "with optimal combination of different cloud tests and smoke detection. . ." Please clarify if you mean A) MAIAC automatically determines the optimal combination of cloud tests and smoke detection; B) The authors have empirically determine the optimal combination and implemented this in MAIAC; C) Within the MAIAC framework, selection of an optimal combination is possible (similar to how the MISR research retrieval can be hand-tuned to obtain the best answer for a given scene)

13-26 "The land surface is considerably brighter at 2.13um compared to the Blue wavelength. This results in spectral dependence of the BRDF shape and in SRC dependence on the view geometry." This demands an additional sentence of explanation.

6-17 This could be clearer. If I understand correctly, LUTs are generated for P=1 for all wavelengths. For wavelengths shorter than 660nm, a second calculation is done for P=0.7. Is this because the pressure correction is largely the signal of the molecular atmosphere, and thus diminishes at longer wavelengths?

5-11 Either here or at the beginning of Section 2, please enumerate the static data used by MAIAC, so that the reader clearly understands that all other values are dynamically updated by the MAIAC processing.

26-9 "seasonal variation in aerosol properties. . ." Do you mean like the variation diagnosed in Eck et al. 2013 (https://doi.org/10.1002/jgrd.50500)?

7-5 It would be nice to have a table tabulating the dimensions of the LUT. If you can do

it in a sentence, that is also OK.

24-21 Give yourself some credit! "The QA structure may be updated in future releases to better accommodate the needs of end users."

23-23 "Following the Sun-View Geometry suite at 5km..." I don't know what you are referring to. Is this another MODIS product?

22-24 Is it the first 16 orbits, or is it the 16 orbits with the largest coverage of the tile?

TYPOS and very minor edits: Page-Line

26-23 Devadiga

25-22 Taklamakan

24-10 most pixels

14-7 Uses the LER surface model

13-31 linear interpolation... is used within 0.01 of bin boundaries

9-1 The test uses the shortest wavelength MODIS channel

9-24 Please be specific about what "neighboring" means here

8-6 This is a good place for a call-forward "See Section 6.2 and 6.3"

4-3 This is a good place for a call-forward (see Test C.4, Section 4.1)

4-15 for most smoke plumes

3-24 For every observation

2-13 we have significantly changed

1-9 "adapt to global processing" or "adopt global processing"

---

## Referee Comment (RC2) · J. Reid (Referee) · 22 Jul 2018

Overview: This is a very well written theoretical basis document sort of paper on the MAIAC algorithm. Often papers are criticized for being too long or involved, but typically I disagree-wanting to know how to reproduce the work and try and get a feel of what the authors are really thinking. This current paper is an outstanding example of how such ATBD papers should be written and published in the peer reviewed literature. I have been familiar with the basic nature of MAIAC, and yet after reading the paper I have an even more thorough understanding and know how to apply the data. The authors discuss the algorithm thoroughly, what works, and what the limitations are. In the "pre review" gave Lyapustin et al., some small corrections that need not be repeated here. I did have a few major comments that I said did not to be address immediately for public review but did need attention before final publication. They are neither hard nor time consuming, but it does I think need to get done. Thus, I would like the following

three things to be addressed in the final revised paper. 1) Prognostic error modeling: As is typical for satellite aerosol products, MAIAC is compared in bulk to all available AERONET in a regression (Figure 7). Bias and RMSE are calculated. I consider this a "diagnostic error model." Now I understand that this paper is more of an ATBD style, and expect that a thorough verification paper will be forthcoming. But one minor addition to Figure 7 (say a Figure 7 b) is a plot of RMSE and or RMSD (if there is a consistent bias) as a function of retrieved (e.g., MAIAC) AOD. That will give you the actual error bar for any retrieval. Consider- if we knew AERONET AOD, we would have no need of MAIAC. You need to know what the error is at any given place and time, and you only have the MAIAC algorithm, to go on. What the authors will undoubtedly find is that their error is parabolic, with a well-defined flat noise floor at low AODs, then growing nonlinearly. I would like to know based on the bulk analysis what that noise floor is, and how nonlinear the error is. The authors can devise more complicated error model, including the addition of other non-MAIAC data at hand, but that for certain is not necessary here.

2) Use for data assimilation: Currently there are nine centers doing operational or near real time global aerosol modeling, four of which with operational data assimilation. Without a doubt, the field is moving towards data assimilation as one of if not the biggest "power users" of retrievals such as MAIAC. In DA, we are often in the situation of violating the fundamental assumption that errors between adjacent retrievals are uncorrelated. I think it should be emphasized in the paper that the way the retrieval has regional optical models outlined in blocks leads to very sharp inconsistencies in AOD across arbitrary lines. Similarly, you can see very strong deviations at the coastlines from time to time.This can be a problem for data assimilation or anyone trying to invert sources and sinks out of the data. This issue is noted in the paper (and I commend the authors for stating what does not work so well), but I think an example figure as to how big a deviation there can actually be.

3) Overlap uncertainty: One thing we noticed when looking at the MAIAC data online,

which I discussed in person with Alexei Lyapustin a few weeks ago was that it appears that under some circumstances the overlap regions between successive orbits can have led to significantly different values in AOD in the overlap. This behavior was inconsistent, being difficult to notice in retrievals of the environment 5 year ago, but being very prominent in data being collected right now. I think this issue should be listed in the paper, and the potential users should be explained what the current best idea of what this is, and any advice as to how we can control for it. Also note the timeline for correction.

For example regarding 2& 3, today as I write this review you can see the sharp change along the meridian in western Africa https://landweb.modaps.eosdis.nasa.gov/browse/images/006/Both/MCD19A2-AOT/2018/A2018191/MCD19A2-AOT.A2018191.006.full.png. You can also see optical model change from land to water and scan lines as well. These spatially correlated errors might need a paragraph of discussion or two. But they are pretty common-just pick any day in the past few years and you will see them popping up. https://landweb.modaps.eosdis.nasa.gov/cgi-bin/browse/browseMODIS.cgi

[Figure]

**Fig. 1.**

---

## Author Comment (AC1) · 17 Aug 2018

- We truly appreciate a very careful and detailed analysis of the Reviewer. The response to the comments is provided below. We accepted all edits and suggestions except one minor case where we feel the information is already provided and repetition would be redundant.

Dear Authors, First, thank you for a well-written detailed manuscript. The MAIAC product you describe is a very significant addition to the Earth system monitoring capability enabled by the MODIS instruments. I have only a few clarifying areas and this manuscript can be ready for publication. I have numerous smaller suggested edits below, and then several typographic edits at the bottom.

[Figure]

1) You should be congratulated for the "Known Issues and Limitations" section, this is very valuable information to compile and include. No one knows this product better than the authors, and this information is very valuable to end users even though it is not fully digested (if it were fully digested it would probably be solved!)

- Thank you for your comment. This section should help users, especially the new ones, in their application or science analysis based on MAIAC dataset.

2) The assumed background AOD seems very important. If the assumed background is too low by 0.01, what is the resulting error in the retrieved AOD?

- In general, at low AOD it will translate into a similar low retrieval bias of ∼0.01. At higher AOD, the uncertainty in aerosol model will play a much larger role than this source of uncertainty.

3) If the "reference SRC" is updated before retrieval of AOD, this means it is impossible for MAIAC to retrieve an AOD lower than the assumed background.

- It is updated after, but in general you are correct. For instance, assuming background AOD∼0.05 works well for the North America in summer, but creates a significant bias in late fall-winter when the background value is closer to ∼0.02. That's why the background AOD in some regions is seasonally-dependent, and it was carefully selected by "calibrating to" the AERONET measurements for each region. However, the retrieved AOD can be lower than the background value due to different sources of errors in MAIAC, some of which can be considered as random (e.g., uncertainties of gridding) or systematic (relatively wide angular bins of SRC which creates albeit small but detectable geometry-dependent bias. For instance, analysis of Superczinsky et al., or the current results of Mhawish et al., indicate the need to add 2 more bins at high VZAs ).

Does the two-stream method described for correcting the "reference SRC" upward mitigate the upward trend in AOD that would be expected from downward updates to the "reference SRC"?

- MAIAC SRC-retrieval procedure was worked out well, and it is quite resistant to noise and errors (e.g., from residual clouds). The major impact on SRC comes from undetected shadows which can reduce the value significantly if undetected. That's why I've put a large effort into a good shadow detection so it won't interfere with SRC retrieval. All other factors (e.g., high AOD or un-detected clouds) tend to increase SRC and don't affect our minimum reflectance method. From this prospective, the main issue is not to go "down" for SRC, but instead is to find the cloud-free day with minimum aerosol for regions with persistent haze like southern China which may require from half a year to a whole year to initialize the SRC.

MINOR notes and edits: Page-Line 19-20 How are cloud shadows flagged in the output product?

- It's in the first 3 bits (0-2) of QA (see Table 2b), value (101 — Cloud Shadow)

4-15 "with optimal combination of different cloud tests and smoke detection:" Please clarify if you mean A) MAIAC automatically determines the optimal combination of cloud tests and smoke detection; B) The authors have empirically determine the optimal combination and implemented this in MAIAC; C) Within the MAIAC framework, selection of an optimal combination is possible (similar to how the MISR research retrieval can be hand-tuned to obtain the best answer for a given scene)

- The answer is B). We changed the relevant sentence as follows: "With optimal combination of different cloud tests and smoke detection, which was found experimentally based on large-scale MODIS data processing, ..."

13-26 "The land surface is considerably brighter at 2.13um compared to the Blue wavelength. This results in spectral dependence of the BRDF shape and in SRC dependence on the view geometry." This demands an additional sentence of explanation.

- The relevant text was changed as follows: "This results in spectral dependence of the BRDF shape: when the surface is dark, the BRDF is well defined by the first order of

scattering whereas in case of bright surface, the photon can experience several scatterings on microfacets of the surface "roughness" before escaping into the atmosphere which results in relative flattening of the BRDF shape. For this reason, SRC depends on the view geometry."

6-17 This could be clearer. If I understand correctly, LUTs are generated for P=1 for all wavelengths. For wavelengths shorter than 660nm, a second calculation is done for P=0.7. Is this because the pressure correction is largely the signal of the molecular atmosphere, and thus diminishes at longer wavelengths?

- That is correct. To make it more clear, the sentence was changed as follows: "Because Rayleigh optical depth rapidly decreases with wavelength, computations with P=0.7 are done for wavelengths shorter than 0.66um."

5-11 Either here or at the beginning of Section 2, please enumerate the static data used by MAIAC, so that the reader clearly understands that all other values are dynamically updated by the MAIAC processing.

- The following phrase was added at the end of Sec. 2: "As ancillary data, MAIAC uses static DEM, 1km land-water mask for deep and static water, and 6-hour NCEP ozone and wind speed."

26-9 "seasonal variation in aerosol properties: " Do you mean like the variation diagnosed in Eck et al. 2013 (https://doi.org/10.1002/jgrd.50500)?

- Yes – and thank you, I provided a wrong reference which has now been corrected.

7-5 It would be nice to have a table tabulating the dimensions of the LUT. If you can do it in a sentence, that is also OK.

- Implicitly, the dimensions are stated in the second sentence on p. 7: "Finally, LUTs are computed for a relatively sparse angular grid (delta_u=0.05 for the range u=0.4 – 1 (0o - 66.42o), ïA■u0=0.15–1 (0o - 81.37o) and delta_AZ= 9o) and 12 AOD values, {0.05, 0.1, 0.2, 0.3, 0.4, 0.55, 0.75, 1., 1.4, 2.0, 2.8, 4.0} giving the size of 45.7MB per

a regional aerosol model."

Translating into specific numbers gives: 16 Bands x 12 AOD (+ Rayleigh) x 13 (VZA) x 18 (SZA) x 21 (relative AZ), and 2 pressure levels. Because this information is redundant, it is not currently provided. However, if the Reviewer thinks it's important I can easily add it.

24-21 Give yourself some credit! "The QA structure may be updated in future releases to better accommodate the needs of end users."

- I appreciate your comment. I meant to say that we didn't spend much time on QA design, and current structure is not optimal.

23-23 "Following the Sun-View Geometry suite at 5km: : :" I don't know what you are referring to. Is this another MODIS product?

- Thank you for noticing this omission. To explain, the following sentence was added at the end of sec. 10.2.1: "Along with the retrieval results, we also provide the "Sun-View geometry" at 5km resolution which includes cosines of solar and view zenith angle, relative azimuth, and scattering and glint angles which may be required for analysis or applications."

22-24 Is it the first 16 orbits, or is it the 16 orbits with the largest coverage of the tile?

- The second. The relevant sentence was corrected as follows: "At high latitudes, only 16 orbits with largest coverage are reported per day in order to limit the file size."

- Below, we accepted all suggested edits – thank you!

TYPOS and very minor edits: Page-Line 26-23 Devadiga 25-22 Taklamakan 24-10 most pixels 14-7 Uses the LER surface model 13-31 linear interpolation: : : is used within 0.01 of bin boundaries 9-1 The test uses the shortest wavelength MODIS channel 9-24 Please be specific about what "neighboring" means here - The sentence was modified as follows: "This second iteration is applied to pixels which are direct neighbors of the detected clouds."

8-6 This is a good place for a call-forward "See Section 6.2 and 6.3"

- Added.

4-3 This is a good place for a call-forward (see Test C.4, Section 4.1)

- Added (and thank you again - you really delved into the details of the algorithm!!!)

4-15 for most smoke plumes 3-24 For every observation 2-13 we have significantly changed 1-9 "adapt to global processing" or "adopt global processing"

---

## Author Comment (AC2) · 17 Aug 2018

- Dear Jeff,

Thank you the review of our paper. You practical perspective representing both wide range of applications and modeling and data assimilation communities is extremely valuable. To address your requests, we added a new section "Prognostic error modeling", and provided more material related to a) AOD discrepancy on the boundaries of geographic regions, and b) continued calibration degradation of MODIS Terra. Our specific response follows your review items.

Thank you,

Alexei.

 Overview: This is a very well written theoretical

basis document sort of paper on the MAIAC algorithm. Often papers are criticized for being too long or involved, but typically I disagree-wanting to know how to reproduce the work and try and get a feel of what the authors are really thinking. This current paper is an outstanding example of how such ATBD papers should be written and published in the peer reviewed literature. I have been familiar with the basic nature of MAIAC, and yet after reading the paper I have an even more thorough understanding and know how to apply the data. The authors discuss the algorithm thoroughly, what works, and what the limitations are. In the "pre review" gave Lyapustin et al., some small corrections that need not be repeated here. I did have a few major comments that I said did not to be address immediately for public review but did need attention before final publication. They are neither hard nor time consuming, but it does I think need to get done. Thus, I would like the following three things to be addressed in the final revised paper. 1) Prognostic error modeling: As is typical for satellite aerosol products, MAIAC is compared in bulk to all available AERONET in a regression (Figure 7). Bias and RMSE are calculated. I consider this a "diagnostic error model." Now I understand that this paper is more of an ATBD style, and expect that a thorough verification paper will be forthcoming. But one minor addition to Figure 7 (say a Figure 7 b) is a plot of RMSE and or RMSD (if there is a consistent bias) as a function of retrieved (e.g., MAIAC) AOD. That will give you the actual error bar for any retrieval. Consider- if we knew AERONET AOD, we would have no need of MAIAC. You need to know what the error is at any given place and time, and you only have the MAIAC algorithm, to go on. What the authors will undoubtedly find is that their error is parabolic, with a well-defined flat noise floor at low AODs, then growing nonlinearly. I would like to know based on the bulk analysis what that noise floor is, and how nonlinear the error is. The authors can devise more complicated error model, including the addition of other non-MAIAC data at hand, but that for certain is not necessary here.

- To accommodate this request, a new section (10.5 Accuracy Assessment of MAIAC AOD) was added with new Figure 8 and Table 5. The new text starts at the second half of this section.

10.5 Accuracy Assessment of MAIAC AOD Figure 7 presents results of the global MAIAC AOD validation against AERONET (Holben et al., 1998) showing correlation coefficient, average bias, and rmse for individual AERONET sites along with the global scatterplot during 2000-2016. The detailed validation analysis of MAIAC dataset, and its comparison with the standard products from MODIS or other sensors deserves a separate consideration, so this analysis merely serves to illustrate the overall quality of MAIAC aerosol retrievals. Figure 7 shows a) predominantly high correlation with AERONET except for the world regions where typically both AOD and its range of variation are low (e.g., south-western USA or south of South American continent); b) globally low bias and rmse except major biomass burning, industrial or mineral dust source regions such as Sahara, Sahel and sub-tropical Africa, Indo-Gangetic Plane, south Asia and China. The higher rmse in these source regions is typical of all aerosol retrieval products and is expected due to high variability of aerosol types and properties, often in combination with the bright land surface increasing uncertainties of satellite retrievals. The bias shows clustering of results, and gives a clear indication for the required tuning of MAIAC regional aerosol models, e.g. in South Asia and China. Some of these biases come from the seasonal variation in aerosol properties (e.g., Mhawish et al., 2018) which will be implemented in the next version of MAIAC. The global scatterplot of Fig. 7 shows that 66% of retrievals (grey area) agree with AERONET within $\pm0.05\pm0.1$AOD which improves over the standard accuracy assessment of 15% from the DT algorithm over land (e.g., Levy et al., 2013). While the global assessment may serve as a useful indicator of accuracy, the true performance of any algorithm is inherently regional and local, as shown by R/rmse/bias statistics for each AERONET site. To generalize these assessments into regional prognostic error models, we computed rmse and bias (or rmsd) binned to retrieved AOD for different world regions. These results are summarized in Figure 8 where the line shows the mean and shaded area represents $\pm$ one standard deviation. Our analysis and results of independent studies (e.g., Superczynski et al., 2017; Mhawish et al., 2018) show that MAIAC AOD has little dependency on view geometry. Although MAIAC accuracy somewhat decreases over

bright surfaces, here the regional analysis was done for all AERONET sites together. Figure 8 shows that the linear model for both mean and standard deviation can serve as a reasonable proxy for both rmse and bias (rmsd), for instance: rmse = a + b×AOD $\pm$ ($\alpha$ + $\beta$×AOD). (23) The regional linear regression model parameters are given in Table 5. This table also includes Australian continent not shown in Figure 8 for the lack of space. A more detailed MAIAC AOD error analysis, as in Sayer et al., (2013), will be given separately.

Table 5. Regional linear regression model parameters for the expected error (rmse) and bias (rmsd).

Figure 8. Absolute error (rmse, top row) and bias (rmsd, bottom row) of MAIAC AOD for different world regions as a function of retrieved AOD. NA and SA stand for the North and South America, respectively.

2) Use for data assimilation: Currently there are nine centers doing operational or near real time global aerosol modeling, four of which with operational data assimilation. Without a doubt, the field is moving towards data assimilation as one of if not the biggest "power users" of retrievals such as MAIAC. In DA, we are often in the situation of violating the fundamental assumption that errors between adjacent retrievals are uncorrelated. I think it should be emphasized in the paper that the way the retrieval has regional optical models outlined in blocks leads to very sharp inconsistencies in AOD across arbitrary lines. Similarly, you can see very strong deviations at the coastlines from time to time. This can be a problem for data assimilation or anyone trying to invert sources and sinks out of the data. This issue is noted in the paper (and I commend the authors for stating what does not work so well), but I think an example figure as to how big a deviation there can actually be.

- Section "Known Issues and Limitations, Item 4, was expanded as follows, with new Figure 9 illustrating mentioned geographic boundaries:

4. "Geographic AOD boundaries may sometimes be observed on borders of the regional aerosol models when they have a significant difference in absorption. While this is not an issue over most of the globe, three transition zones may stand out during the biomass burning seasons (see Figure 4): the north-west boundary between India (model 8) and central Asia (model 2), and two transitions from central Africa to Sahel-Sahara (models 7-6) and to the southern Africa (models 7-2). Figure 9 shows one the worst case examples for each transition zone when at high AOD the contrast across the model boundary can be as high as 40-50% of the mean value, while it is not noticeable for most of the year when AOD is moderate-low.

Figure 9. Illustration of MAIAC AOD contrast on the boundary of aerosol models (see Fig. 4) caused mainly by the difference in aerosol absorption between the models: (a) transition 8-2 (day 82, 2010); (b) transition 7-6 (day 113, 2010); (c) transition 7-2 (day 237, 2010).

3) Overlap uncertainty: One thing we noticed when looking at the MAIAC data online, which I discussed in person with Alexei Lyapustin a few weeks ago was that it appears that under some circumstances the overlap regions between successive orbits can have led to significantly different values in AOD in the overlap. This behavior was inconsistent, being difficult to notice in retrievals of the environment 5 year ago, but being very prominent in data being collected right now. I think this issue should be listed in the paper, and the potential users should be explained what the current best idea of what this is, and any advice as to how we can control for it. Also note the timeline for correction. For example regarding 2& 3, today as I write this review you can see the sharp change along the meridian in western Africa   https://landweb.modaps.eosdis.nasa.gov/browse/images/006/Both/MCD19A2-AOT/2018/A2018191/MCD19A2-AOT.A2018191.006.full.png. You can also see optical model change from land to water and scan lines as well. These spatially correlated errors might need a paragraph of discussion or two. But they are pretty common-just pick any day in the past few years and you will see them popping up. https://landweb.modaps.eosdis.nasa.gov/cgi-bin/browse/browseMODIS.cgi

- There are two different issues related to the question you raise. The first one is the global browse images of AOD in the MODAPS which are designed to show a qualitative global view of aerosol, but were never optimized for the science use. Currently, the browse algorithm sim-ply takes one of the sensors (Terra or Aqua) as the baseline, and fills-in the gaps (e.g. caused by clouds or glint over water) with the data from MODIS on the other platform (or other orbits disregarding the time difference). Due to the time difference between the orbits, in case of significant aerosol events (like dust storms over Sahara), such a straightforward approach will inevitably create boundaries clearly seen on the shown image. Since the nature of such "boundaries" is generic, one can easily find them for the earlier years before 2014, e.g. DOY 233, 242, 258 of 2010 for the Saharan coast of West Africa (the same tile). At the same time, the data from individual sensors are smooth. For instance, below we show the same image you provided as an example, followed by MAIAC retrievals for 2 tiles h16v06 and h17v06 for MODIS Terra and Aqua separately (indicated by letter A or T at the end of the time stamp provided on the images). The AOD scale for all images is the same, 0-2. For the tile h16v06 we have one orbit from Terra and 2 orbits from Aqua, the last two separated by 1h25m. AOD from Terra at 11:50 and from earlier Aqua 13:30 are consistent, although AOD from the second Aqua overpass is significantly lower over land.

Figure

The second issue, which is indeed important, is related to the calibration updates of MODIS sensors. The current MODIS calibration approach (Lyapustin et al., 2014b) consists of time-dependent 1) response-vs-scan angle (RVS) correction by the MODIS Calibration Support Team (MCST); 2) polarization sensitivity correction of MODIS Terra by the NASA GSFC Ocean Biology Processing group (OBPG), and 3) residual de-trending and Terra to Aqua cross-calibration by my group. Last time, all three were consistently updated in 2013. Since that time, only RVS was updated; as a result, continued MODIS Terra degradation increases in latest years seen as striping and

growing bias on the left-hand side of the scan in AOD images. The latter (bias) shows now consistently at low amplitude mostly over bright surfaces in the combined AOD browse images, which you are mentioning.

-

Some hints of striping are now apparent in MAIAC AOD from MODIS Aqua as well. MCST, OBPG and our group have updated MODIS Terra and Aqua calibration in late 2017-early 2018, and it is now in testing at MODAPS. It will be implemented in MODIS Collection 6.1 Land Discipline re-processing (which includes MAIAC) scheduled for the second half of 2018. We expect a significant reduction of the discussed artifacts in MAIAC C6.1.

The above discussion was summarized and added to the "Known Issues and Limitations" section: Since 2014, when MODIS Terra/Aqua calibration was consistently updated (Lyapustin et al., 2014b), the continued calibration degradation of MODIS Terra increasingly shows in MAIAC AOD as striping artefacts and positive bias at left-hand side of the MODIS scan mostly over bright surfaces. The MODIS calibration was recently updated. It will be implemented in MODIS Collection 6.1 Land Discipline re-processing (which includes MAIAC) scheduled for the second half of 2018. We expect a significant reduction of mentioned errors in MAIAC C6.1 AOD.

Please also note the supplement to this comment:
https://www.atmos-meas-tech-discuss.net/amt-2018-141/amt-2018-141-AC2-supplement.pdf

[Figure]

**Figure 8.** Absolute error (*rmse*, top row) and bias (*rmsd*, bottom row) of MAIAC AOD for different world regions as a function of retrieved AOD. NA and SA stand for the North and South America, respectively.

**Fig. 1.**

**Figure 9**. Illustration of MAIAC AOD contrast on the boundary of aerosol models (see Fig. 4) caused mainly by the difference in aerosol absorption between the models: (a) transition 8-2 (day 82, 2010); (b) transition 7-6 (day 113, 2010); (c) transition 7-2 (day 237, 2010).

**Fig. 2.**

| Regions | rmse | | $\sigma_{rmse}$ | |
|---|---|---|---|---|
| | a | b | α | β |
| NA | 0.034 | 0.13 | 0.049 | 0.20 |
| SA | 0.049 | 0.063 | 0.083 | 0.041 |
| Asia | 0.057 | 0.13 | 0.083 | 0.12 |
| Europe | 0.035 | 0.15 | 0.055 | 0.18 |
| Africa | 0.049 | 0.21 | 0.087 | 0.17 |
| Australia | 0.05 | 0.088 | 0.094 | 0.08 |
| Regions | rmsd | | $\sigma_{rmsd}$ | |
| | a | b | α | β |
| NA | -0.0081 | -0.0034 | 0.031 | 0.31 |
| SA | -0.017 | 0.0065 | 0.11 | 0.07 |
| Asia | -0.040 | 0.067 | 0.092 | 0.18 |
| Europe | -0.019 | -0.021 | 0.031 | 0.33 |
| Africa | -0.0056 | -0.028 | 0.088 | 0.30 |
| Australia | -0.0076 | 0.085 | 0.15 | 0.11 |

**Table 5.** Regional linear regression model parameters for the expected error (*rmse*) and bias (*rmsd*).

**Fig. 3.**

[Figure]

**Fig. 4.**

---

## Author Response (AR2)

Review by Jeff Reid:

"One mislabeling in the paper worth noting. For figure 8 they list rmsd and bais as the same thing, when in fact they are polar opposites. RMSD (or root mean square deviation) is essentially the rmse one the bias is removed. RMSE is the sqrt of bias^2 plus variance squared. Both rmse and rmsd are positive definite. If they want to show bias instead of rmsd (as I suggested) that is fine. Just relabel figure 8."

Thank you Jeff,

That's correct, what we are showing is a "bias". We removed all references to "rmsd" in the text and relabled Fig. 8.

Specific changes:

P25, Ln. 30:

Was: "prognostic error models, we computed *rmse* and *bias* (or *rmsd*) ..."

Now: "prognostic error models, we computed *rmse* and *bias* ...."

P26, Ln. 2:

Was: "deviation can serve as a reasonable proxy for both *rmse* and bias (*rmsd*), for instance: "

Now: "deviation can serve as a reasonable proxy for both *rmse* and bias, for instance: "

P. 26, Ln 4-5:  Removed the sentence:

"This table also includes Australian continent not shown in Figure 8 for the lack of space."

because Fig. 8 was updated with Australia added.

P35, Ln. 18 (and P42):

Was: " **Table 5.** Regional linear regression model parameters for the expected error (*rmse*) and bias (*rmsd*)."

Now: "**Table 5.** Regional linear regression model parameters for the expected error (*rmse*) and bias."

P43, Ln. 26 (and P. 52):

Was: "**Figure 8.** Absolute error (*rmse*, top row) and bias (*rmsd*, bottom row) "

Now: "**Figure 8.** Bias (top row) and *rmse* (bottom row) ..."

Updated Fig. 8:

1) Replaced "RMSD" with "Bias"
2) Re-plotted $RMSE$ with y-axis starting from 0 to address Reviewer's comment that "$RMSE$ is a positively defined function".
3) Added Australian continent.